# Kalman Linear Attention: Parallel Bayesian Filtering For Efficient Language Modelling and State Tracking

Vaisakh Shaj [1]   Cameron Barker [1]   Aidan Scannell [1]   Andras Szecsenyi [1]   Elliot J. Crowley [1]   Amos Storkey [1]

## Abstract

State-space language models such as Mamba and gated linear attention (GLA) offer linear-complexity, parallelisable alternatives to transformers, but often lack the expressivity and robust state-tracking needed for complex reasoning. We address this by reframing sequence modelling through a probabilistic lens, with Bayesian filters as the core primitive. Classical Kalman filters provide principled state and uncertainty estimation but are typically viewed as inherently sequential. We show that reparameterising the Kalman filter in information form casts its updates as an associative scan, enabling efficient parallel training. The resulting Kalman Linear Attention (KLA) layer performs time-parallel probabilistic inference while maintaining explicit belief-state uncertainty, offering strictly more expressive nonlinear updates and gating than GLA variants while retaining their computational advantages. On language modelling tasks, KLA matches or outperforms modern SSMs and GLAs on discrete token-manipulation, state-tracking, and zero-shot commonsense-reasoning benchmarks, and is among the first stacked Bayesian-filtering primitives pretrained at the billion-token scale.

## 1. Introduction

Scaling sequence modelling beyond quadratic attention (Vaswani et al., 2017) is a central goal in large language and foundation model research.[1] This has motivated interest in RNN-like architectures that support linear-time sequence modelling with efficient parallel training. Recent state-space models (SSMs), including S4/S5 (Gu et al., 2021; Smith et al.,

| | Softmax Attention | SSMs / GLA | KLA |
|---|---|---|---|
| Expressivity | Nonlinear | Linear | Fractional lin. (Möbius) |
| Training eff. | $\mathcal{O}(T^2)$ | $\mathcal{O}(T)$ | $\mathcal{O}(T)$ |
| Inference eff. | $\mathcal{O}(T)$ | $\mathcal{O}(1)$ | $\mathcal{O}(1)$ |
| Seq. uncertainty | ✗ | ✗ | ✓ |
| Parallel training | ✓ | ✓ | ✓ |

*Table 1.* High-level comparison of sequence-mixing primitives.

2022), Mamba (Gu & Dao, 2023; Dao & Gu, 2024), and their successors, achieve this with $\mathcal{O}(\log T)$ depth, $\mathcal{O}(T)$ work, and linear or sublinear memory. This is essential for handling long contexts, on-device deployment and energy efficiency.

Despite their efficiency, adoption of SSMs is not as widespread as attention-based transformers. Recent work reinterprets modern SSMs and linear RNNs, including Mamba, as gated variants of linear attention (GLA) (Yang et al., 2023), where performance largely depends on how gates are defined. In all of these, the underlying hidden state updates are linear or affine, which limits their expressivity compared to softmax attention, whose normalisation induces nonlinear interactions between tokens. Furthermore, none of these represent state uncertainty explicitly. This contrasts with longstanding probabilistic linear state-space formalisms that lead to Kalman filter updates. These have both state uncertainty and nonlinear parameter updates.

In this paper, we ask: *Can we overcome the linear update constraints common to current GLA models to develop a state-space block that efficiently implements Kalman-like updates?*

We introduce **Kalman Linear Attention (KLA)**, which formulates sequence modelling as a Bayesian filtering problem. KLA models two sources of uncertainty: process noise, which captures uncertainty in state evolution, and observation noise, which captures uncertainty in the information provided by each input token. Crucially, this uncertainty is not merely an output: it directly controls how new information is gated in state spaces.

At first glance, such Bayesian updates seem inherently sequential. Our key insight is that the information form of the Kalman filter admits a fractional-linear / Möbius associative structure that can be implemented using parallel scan-based algorithms, *despite its nonlinear update computation*. This allows Kalman-style updates to achieve the same $\mathcal{O}(\log T)$

[1]University of Edinburgh, Edinburgh, United Kingdom. Correspondence to: Vaisakh Shaj <vaisakhs.shaj@gmail.com>.

*Proceedings of the $43^{rd}$ International Conference on Machine Learning*, Seoul, South Korea. PMLR 306, 2026. Copyright 2026 by the author(s).

[1]Code: github.com/vaisakh-shaj/kalman-linear-attention

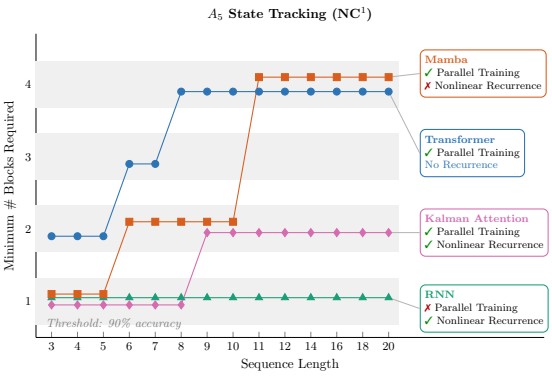

*Figure 1.* **(a) Expressivity.** Minimum number of blocks required to solve the $A_5$ (alternating group on 5 elements) permutation-composition task (Merrill et al., 2024). KLA's fractional-linear (Möbius) updates fall between a fully nonlinear RNN and linear SSMs/transformers, solving the task at constant depth (1-2 blocks) where linear SSMs/attention require depth growing with sequence length, while remaining parallel-trainable. **(b) Downstream scaling.** Zero-shot accuracy (averaged over eight commonsense benchmarks) of softmax-attention GPT vs. GPT+SSM hybrids at 45M and 180M parameters. Replacing a single final attention layer with KLA (GPT+KLA) gives the strongest complement to attention.

parallel depth as models such as Mamba. The result is the **KLA** layer: a drop-in replacement for standard SSM or attention layers.

It is not enough to implement a new primitive without establishing that the extra expressiveness provides a benefit. We study the suitability of this continuous-state probabilistic model for discrete language modelling both theoretically and empirically, demonstrating important gains.

**Contributions.** **(C1) Associative reparameterisation of Kalman filtering:** we reparameterise the diagonal linear-Gaussian filter in information form, showing the precision recursion is a Möbius (fractional-linear) map that composes associatively, enabling parallel prefix scans. **(C2) Nonlinear gating from uncertainty:** the precision-ratio gates are history-dependent and nonlinear, going beyond linear/affine SSM and gated linear-attention updates while preserving linear-time scan structure. **(C3) Kalman Linear Attention layer:** we introduce KLA, a drop-in sequence mixer for modern language-modelling pipelines that produces explicit belief-state uncertainty. **(C4) Scaling and empirical validation:** our scan-based implementation scales efficiently with sequence length and matches or outperforms modern SSMs and GLAs on state-tracking, associative recall, and zero-shot commonsense reasoning, and is among the first stacked Bayesian filters pretrained at the billion-token scale.

**Conflict of Interest Disclosure.** The authors declare no financial or other conflicts of interest; this is purely academic work.

## 2. Related Work

**Attention as filtering.** The idea that intelligent systems must filter information predates modern self-attention: in cognitive science, early theories framed attention as a mechanism that suppresses irrelevant signals and routes task-relevant ones under limited processing capacity (Broadbent, 1957; Treisman, 1969; Deutsch & Deutsch, 1963; Nielsen & Garcia, 2009). The core theme across these views is that attention allocates scarce processing by selecting what matters in context.

In modern language models, transformer self-attention can be viewed as a form of context-based filtering (Vaswani et al., 2017). Tokens compete via similarity and the softmax suppresses most inputs whilst allowing a small subset to dominate the update. But this selection is *exemplar-based*: it requires retaining and comparing against all past keys/values, largely eliminating an explicit bottleneck and yielding $\mathcal{O}(T^2)$ cost at long context lengths.

**Bayesian filtering and precision-control as attention.** Bayesian filters perform filtering as posterior inference: prior beliefs are updated with new observations weighted by their precision (inverse variance), so more reliable observations exert greater influence whilst unreliable evidence is downweighted. This yields a principled, interpretable gating mechanism where selection follows from uncertainty propagation.

Precision-weighted prediction errors have also been proposed as a computational account of attention in neuroscience and predictive coding (Rao & Ballard, 1999; Feldman & Friston, 2010). In this view, attention modulation is based upon confidence in sensory signals as opposed to deterministic reweighting. We adopt this view and show that Bayesian filters can be integrated into modern sequence modelling.

**Sequence models as test-time in-context online learners.** Concurrently, recent sub-quadratic mixers have been unified as online learners that solve a per-token regularised regression in closed form, a view first articulated for SSMs by Longhorn (Liu et al., 2025; Wang et al., 2025) and extended in linear-attention and delta-rule variants (Yang et al., 2023;

2024; von Oswald et al., 2025; Hatamizadeh et al., 2026); we situate KLA within this view in Section 4.2 (with the full template in Section E), where its posterior-mean update emerges as a gated RNN update with an adaptive learning rate from filtering.

**Temporal parallelisation of Bayesian filters.** Särkkä & García-Fernández (2021) pioneered temporal parallelisation of Bayesian filtering and smoothing using prefix-sum algorithms, requiring specialised parameterisations to ensure associativity. To achieve associativity, they construct it by lifting each filtering step into a 5-tuple augmented representation, in the linear case. We show that no such augmentation is needed: Kalman updates in the information form are already Möbius maps, so precision composes by $2 \times 2$ matrix multiplication, a minimal, GPU-friendly structure (Theorems 1 and 2). Beyond parallelisation, we integrate these probabilistic primitives into learnable neural sequence models for language, where token-dependent observation likelihoods and dynamics are learned from data rather than specified a priori.

**Deep Kalman filters and probabilistic state-space models.** Kalman filtering (Kalman, 1960) has a long history in RL, control, and world modelling (Haarnoja et al., 2017; Watter et al., 2015; Hafner et al., 2019; 2020; Shaj et al., 2023). Deep Kalman filters embed linear-Gaussian SSMs in neural networks, trained by exact or variational inference (Krishnan et al., 2015; 2017; Karl et al., 2017; Fraccaro et al., 2016; Becker et al., 2019; Shaj et al., 2021), but their recurrent update limits scalability to long contexts. Recent deep-RL work pairs deterministic SSM backbones with Kalman-style dynamics via Särkkä's temporal parallelisation (Becker et al., 2024), or replaces them with parallelised Kalman layers for uncertainty under partial observability (Luis et al., 2024). In contrast, we adopt KLA as a *standalone*, closed-form probabilistic primitive for language modelling, competing directly with modern SSM/GLA mixers without an extra deterministic scaffold. Crucially, whereas these prior approaches typically employ a single Kalman layer, we hierarchically stack KLA across many layers, training deep multi-layer Bayesian-filtering architectures for language modelling at the billion-token scale.

## 3. Background and Preliminaries

Modern language models compose a shared representation map $f(\cdot; \theta)$ and a task-specific head $g$ (Alonso et al., 2025): given a prompt $x_{1:T}$, the backbone produces latents $z_{1:T} = f(x_{1:T}; \theta)$ and the head maps them to outputs $y_{1:T}$ (e.g., next-token prediction, tagging, regression). Concrete parameterisations for $f$ balance expressivity with computational tractability: transformers use softmax attention as the core primitive, whereas state-space models (SSMs) such as

Mamba adopt structured recurrence with linear complexity.

**Deterministic state-space models (SSMs) and Mamba.** Popular SSMs such as Mamba originate from structured state-space models (Gu et al., 2021). Building on their diagonal simplification (Gupta et al., 2022), a linear SSM maps a scalar input signal $x(t)$ to an output $y(t)$ through a latent state $h(t) \in \mathbb{R}^N$:

$$h'(t) = \mathbf{A}h(t) + \mathbf{B}x(t), \qquad y(t) = \mathbf{C}h(t), \qquad (1)$$

with $x(t), y(t) \in \mathbb{R}$, $\mathbf{A} = \mathrm{diag}(a_1, \ldots, a_N)$, $\mathbf{B} \in \mathbb{R}^N$, and $\mathbf{C} \in \mathbb{R}^N$. After discretisation, one processes a $D$-dimensional sequence $\mathbf{X} \in \mathbb{R}^{T \times D}$ by running $D$ independent copies of this recurrence, one per feature coordinate, or channel-, each with its own scalar input $x(t)$.

**Mamba (S6)** (Gu & Dao, 2023) makes this recurrence input-dependent (selective). For a single channel the update is

$$\mathbf{h}_t = \bar{\mathbf{a}}_t \odot \mathbf{h}_{t-1} + \bar{\mathbf{b}}_t\, x_t, \qquad y_t = \mathbf{c}_t^\top \mathbf{h}_t, \qquad (2)$$

where $x_t, y_t \in \mathbb{R}$; this system can be applied independently to each feature channel of a $D$-dimensional sequence. Here $\mathbf{h}_t \in \mathbb{R}^N$ is the latent state, $\bar{\mathbf{a}}_t, \bar{\mathbf{b}}_t \in \mathbb{R}^N$ are discretised diagonal dynamics, and $\mathbf{c}_t \in \mathbb{R}^N$ is an input-dependent readout (the selective counterpart of $\mathbf{C}$ in Equation (1)). The key property is *selectivity*: all parameters $(\bar{\mathbf{a}}_t, \bar{\mathbf{b}}_t, \mathbf{c}_t)$ are computed from the current input $x_t$, allowing the model to selectively filter which information to retain or discard at each step, while preserving linear-time scanning. This input-dependent gating can be seen as a learned form of filtering; in Section 3 we formalise this intuition through Bayesian filtering with explicit uncertainty.

**State expansion.** The dimension $N$ in Equation (2) is a design choice known as *state expansion* (Gupta et al., 2022; Poli et al., 2024; Yu & Erichson, 2025). With $N{=}1$, each feature channel maintains a single scalar recurrence and the state is $\mathbf{h}_t \in \mathbb{R}^D$. Setting $N{>}1$ equips every channel with $N$ parallel memory slots, expanding the state to $\mathbf{H}_t \in \mathbb{R}^{N \times D}$. Stacking the per-channel dynamics accordingly, we write $\bar{\mathbf{A}}_t, \bar{\mathbf{B}}_t \in \mathbb{R}^{N \times D}$ and $\mathbf{C}_t \in \mathbb{R}^{1 \times N}$. How the dynamics are allocated across these slots is itself an algorithmic decision: $\bar{\mathbf{A}}_t$ and $\bar{\mathbf{B}}_t$ may be *distinct* per slot or *shared* (tied) across channels, depending on the model variant. State expansion trades a larger memory footprint ($N \times D$ vs. $D$) for richer per-channel recurrence histories, as each slot can capture a different effective time-scale.

**Attention from the SSM perspective.** Ignoring scaling constants, auto-regressive softmax self-attention computes

$$\mathbf{Y} = \mathrm{softmax}(\mathbf{Q}\mathbf{K}^\top + \mathbf{M})\mathbf{V}, \qquad (3)$$

where $\mathbf{Q}, \mathbf{K}, \mathbf{V} = \mathbf{X}\mathbf{W}_q, \mathbf{X}\mathbf{W}_k, \mathbf{X}\mathbf{W}_v$ for $\mathbf{X}, \mathbf{Q}, \mathbf{K}, \mathbf{V} \in \mathbb{R}^{T \times D}$, $\mathbf{W}_q, \mathbf{W}_k, \mathbf{W}_v \in \mathbb{R}^{D \times D}$ and $\mathbf{M} \in (-\infty, 1)^{T \times T}$ is an auto-regressive mask.

Unlike the linear recurrence in Equation (2), softmax attention maintains a distinct memory of each $\mathbf{K}_t$ and $\mathbf{V}_t$, yielding $\mathcal{O}(T^2)$ time and memory complexity.

By removing softmax we obtain Linear Attention $\mathbf{y}_t = \mathbf{q}_t \sum_{i=0}^{t} \mathbf{k}_i^\top \mathbf{v}_i$ which permits an alternate recurrent form (Katharopoulos et al., 2020) that can be calculated in $\mathcal{O}(T)$ time and $\mathcal{O}(1)$ memory complexity, given by

$$\mathbf{H}_t = \mathbf{G}_t \odot \mathbf{H}_{t-1} + \mathbf{k}_t^\top \mathbf{v}_t \qquad \text{and} \qquad \mathbf{y}_t = \mathbf{q}_t \mathbf{H}_t. \quad (4)$$

where $\mathbf{G}_t \in \mathbb{R}^{N \times D}$ is an optional gating term.

Identifying $\mathbf{G}_t \equiv \bar{\mathbf{A}}_t$, $\mathbf{k}_t^\top \equiv \bar{\mathbf{B}}_t$ and $\mathbf{q}_t \equiv \mathbf{C}_t$ shows that GLA (Yang et al., 2023) matches Mamba when $\mathbf{W}_v = \mathbf{I}$.

**Gaussian state-space models and Bayesian filtering.** A classical probabilistic lens on sequential data is *state estimation*: we posit an (unobserved) latent state $\mathbf{z}_t \in \mathbb{R}^d$ that evolves over time, and assume the observed sequence $\mathbf{o}_t \in \mathbb{R}^p$ provides noisy evidence about that state. The goal of *Bayesian filtering* is to maintain the posterior belief $p(\mathbf{z}_t \mid \mathbf{o}_{1:t})$ online, recursively combining a *dynamics prior* (predict) with an *observation likelihood* (update).

When the dynamics and observations are linear with Gaussian noise, we obtain a linear-Gaussian state-space model:

$$\mathbf{z}_t = \mathbf{A}_t \mathbf{z}_{t-1} + \mathbf{w}_t, \qquad \mathbf{w}_t \sim \mathcal{N}(\mathbf{0}, \mathbf{Q}_t), \quad (5)$$

$$\mathbf{o}_t = \mathbf{C}_t \mathbf{z}_t + \mathbf{v}_t, \qquad \mathbf{v}_t \sim \mathcal{N}(\mathbf{0}, \mathbf{\Sigma}_t^{\text{obs}}), \quad (6)$$

in which the filtering posterior remains Gaussian, $p(\mathbf{z}_t \mid \mathbf{o}_{1:t}) = \mathcal{N}(\boldsymbol{\mu}_t, \mathbf{\Sigma}_t)$ and can be computed in closed form via the *Kalman filter* (KF), which is optimal in the minimum mean-squared error (MMSE) sense under these assumptions.

**Control versus observation: two ways to read a token sequence.** This filtering view contrasts with the common "controlled dynamics" interpretation of deterministic mixers (e.g. SSM/CDE-style models) (Muca Cirone et al., 2024), where the input is an exogenous control driving the hidden state forward; Bayesian filtering instead treats it as noisy observations of a stochastic latent process. This swap, control versus observation, changes what is random and what computation is appropriate, forward simulation versus posterior inference (Figure 6). We adopt the filtering perspective throughout, yielding an attention-like mixer with explicit uncertainty propagation.

**Notational conventions.** We index token sequences by $t \in \{1, \ldots, T\}$. Bold uppercase terms denote matrices (or collections over time), lowercase denotes scalars, and boldface denotes vectors/sequences (e.g., $\mathbf{x}_t$). In the diagonal (per-coordinate) parameterisation, we identify diagonal matrices with their diagonal terms and apply scalar recursions elementwise. An overbar denotes the discretised counterpart of a continuous-time parameter, e.g. $\bar{a}_t$ is obtained by discretising a continuous-time decay parameter.

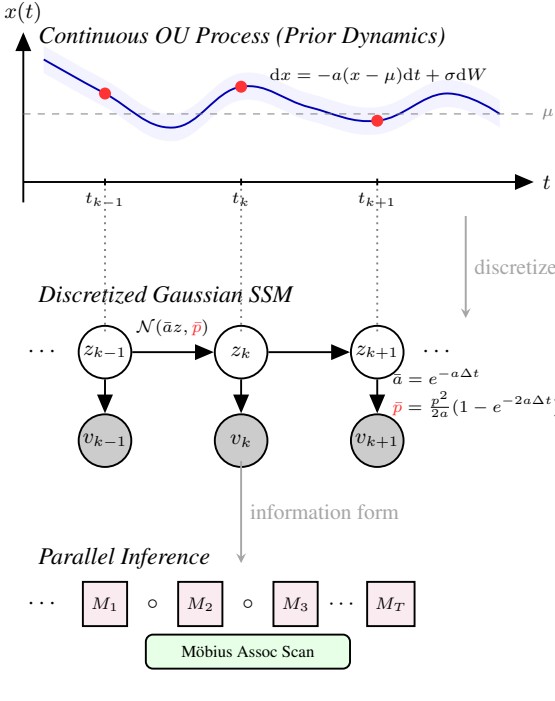

*Figure 2.* From OU dynamics to parallel inference. **(Top)** Continuous-time OU prior. **(Middle)** Discrete linear-Gaussian SSM. **(Bottom)** Möbius scan for parallel posterior state estimation.

For completeness, Appendix D includes additional background material on related mathematical constructs such as Möbius (Fractional Linear) Transforms, the Information Form of Gaussian distributions, and Information Filters.

## 4. Method

We introduce *Kalman Linear Attention (KLA)* as a *probabilistic* sequence mixer. Instead of deterministically updating a hidden state as in modern SSM mixers, KLA maintains a belief state over a latent representation, consisting of a posterior mean and an explicit uncertainty (precision/covariance). This probabilistic view turns sequence mixing into state estimation from noisy token evidence, and provides a principled mechanism for adaptivity and a gating mechanism derived from relative uncertainties of state and observation.

Our method is organised around two ideas. **(i) A probabilistic SSM view of sequence mixing.** We cast token processing as inference in a linear-Gaussian state-space model. Each token contributes a (noisy) observation $v_t$ of the latent state through an observation operator $k_t$, while the latent evolves under a continuous-time stochastic prior that we discretise for sequences.

**(ii) Parallelisable filtering via information form.** We address the main obstacle to using Bayesian filters as sequence mixers: classical filtering is recursive and appears inherently sequential. We show that by working in *information form*

(precisions and natural parameters), the posterior updates acquire a parallelisation-friendly Möbius (fractional-linear) structure. This yields a scalable implementation of Bayesian sequence mixing with the computational profile of modern linear-time mixers, while preserving clear probabilistic semantics.

**Notation and scope.** Throughout, blue terms denote uncertainty-aware quantities absent in deterministic SSMs. For readability, the main text fixes the state-expansion factor to $N{=}1$ and presents the *diagonal* (per-channel) Kalman filter, so that every recursion reduces to scalar (elementwise) operations; the state-expanded ($N{>}1$) matrix form generalises this directly (the state updates of Table 2 and Algorithm 1).[2]

### 4.1. Stochastic Dynamics and Token Likelihood

**Dynamics prior (predict step) via OU discretisation.** Modern SSM mixers (Gu et al., 2021; Smith et al., 2022) are typically motivated from continuous-time dynamics and discretised to operate on tokens: a simple linear ODE yields a discrete recurrence whose decay $\bar{\mathbf{a}}_t$ acts as a forget factor and step size $\Delta t$ sets the time scale, casting SSM channels as a spectrum of memories indexed by decay rate.

KLA keeps this continuous-to-discrete pipeline but makes one change: we treat latent evolution as inherently uncertain, specifying a continuous-time *stochastic* prior and discretising it exactly. We use an Ornstein-Uhlenbeck (OU) process, the canonical mean-reverting diffusion (continuous-time analogue of a stable AR(1)), which preserves exponential forgetting while adding an explicit process-noise term capturing unmodelled variability in the latent dynamics. Under exact discretisation, the OU prior induces a Gaussian transition

$$\mathbf{z}_t \mid \mathbf{z}_{t-1} \sim \mathcal{N}\big(\bar{\mathbf{a}}_t \odot \mathbf{z}_{t-1},\, \bar{\mathbf{p}}_t\big), \qquad (7)$$

$$\bar{\mathbf{a}}_t = e^{-\mathbf{a}\,\Delta t}, \qquad \bar{\mathbf{p}}_t = \frac{\mathbf{p}^2}{2\mathbf{a}} \odot \left(1 - e^{-2\mathbf{a}\,\Delta t}\right), \quad (8)$$

where $\bar{\mathbf{a}}_t$ denotes the decay and $\bar{\mathbf{p}}_t$ the process-noise variance.

**Multi-channel specialisation.** By construction $\bar{\mathbf{p}}_t$ is coupled to $\bar{\mathbf{a}}_t$: the same parameters $\mathbf{a}$ that set decay also govern how uncertainty accumulates between observations, giving per-channel specialisation along two linked axes, memory decay and drift. Different coordinates learn different decay rates (short- vs. long-memory channels),

---

[2]KLA uses two equivalent state parameterisations related by the precision $\mathbf{\Lambda}_t$: the parallel scan is performed in the *information-form* mean $\mathbf{H}_t = \boldsymbol{\eta}_t$ (Algorithm 1), which admits an equivalent *moment-form* representation $\mathbf{S}_t = \mathbf{H}_t \oslash \mathbf{\Lambda}_t$ (i.e. $\boldsymbol{\mu}_t = \boldsymbol{\eta}_t \oslash \boldsymbol{\lambda}_t$) used by the query readout; this moment form has an equivalent recursion as in Table 2. The state-expanded state $\in \mathbb{R}^{N \times D}$ arises *solely* from state expansion (parallel memory slots per channel) and is *not* a full covariance: KLA's covariance/precision remains diagonal throughout.

with process noise $\bar{\mathbf{p}}_t$ setting how far each channel may drift between tokens. Unlike Mamba, where the token-dependent timescale $\Delta_t$ also performs selection/filtering, KLA achieves selection purely via uncertainties without overloading $\Delta_t$.

**Input token likelihood as noisy evidence (update step).** To connect this dynamics prior to input token sequences, we model each token as providing noisy evidence about the latent state:

$$\mathbf{v}_t \mid \mathbf{z}_t \sim \mathcal{N}\big(\mathbf{k}_t \odot \mathbf{z}_t,\, (\mathbf{\Lambda}_t^{\mathrm{v}})^{-1}\big), \qquad (9)$$

where $\mathbf{v}_t$ is token-derived content, $\mathbf{k}_t$ is an observation operator, and $\mathbf{\Lambda}_t^{\mathrm{v}}$ is the value precision representing confidence in the token evidence. Combining Equation (7) with Equation (9) yields a linear-Gaussian state-space model, and *sequence mixing corresponds to posterior inference* of the latent state given token evidence.

**Output readout as a query/task-conditioned projection of the belief state.** Having defined how token evidence updates the belief $p(\mathbf{z}_t \mid \mathbf{v}_{1:t})$, we must read it out for prediction. Just as deterministic SSMs use a $\mathbf{C}_t$-projection $\mathbf{y}_t = \mathbf{C}_t \odot \mathbf{h}_t$ and transformers use a query, KLA's *query* $\mathbf{q}_t$ specifies *what to read out* from the inferred belief at time $t$ via a (linear-Gaussian) readout model

$$\mathbf{y}_t \mid \mathbf{z}_t \sim \mathcal{N}\big(\mathbf{q}_t \odot \mathbf{z}_t,\, (\mathbf{\Lambda}_t^{\mathrm{out}})^{-1}\big), \qquad (10)$$

where $\mathbf{\Lambda}_t^{\mathrm{out}}$ is an *output precision* (readout noise). In this work we take the deterministic-readout limit $\mathbf{\Lambda}_t^{\mathrm{out}} \to \infty$ (zero readout noise), so that the output is the posterior-mean projection $\mathbf{y}_t = \mathbf{q}_t \odot \boldsymbol{\mu}_t$, which mirrors a query-conditioned readout in attention and a $\mathbf{C}_t$-projection in SSMs: beliefs are accumulated from an input modality $\mathbf{v}_t$ via observation model $\mathbf{k}_t$ and read out to $\mathbf{y}_t$ via query model $\mathbf{q}_t$.

**Gaussian SSM view of $q$-$k$-$v$ interactions.** We can summarise the attention-aligned probabilistic interpretation as a linear-Gaussian state-space model in which keys and values parameterise the *likelihood* (token evidence), while queries parameterise a *readout* applied after inference. Let $\mathbf{z}_t$ denote the latent state and let $(\mathbf{k}_t, \mathbf{v}_t, \mathbf{\Lambda}_t^{\mathrm{v}})$ be token-dependent likelihood parameters produced from the input sequence. The generative model is:

| | | |
|---|---|---|
| **Prior** | $\mathbf{z}_t \mid \mathbf{z}_{t-1} \sim \mathcal{N}\big(\bar{\mathbf{a}}_t \odot \mathbf{z}_{t-1},\, \bar{\mathbf{p}}_t\big),$ | (11) |
| **Evidence** | $\mathbf{v}_t \mid \mathbf{z}_t \sim \mathcal{N}\big(\mathbf{k}_t \odot \mathbf{z}_t,\, (\mathbf{\Lambda}_t^{\mathrm{v}})^{-1}\big),$ | (12) |
| **Readout** | $\mathbf{y}_t \mid \mathbf{z}_t \sim \mathcal{N}\big(\mathbf{q}_t \odot \mathbf{z}_t,\, (\mathbf{\Lambda}_t^{\mathrm{out}})^{-1}\big).$ | (13) |

In this view, $\mathbf{k}_t$ specifies the observation geometry (how token evidence constrains the latent), $\mathbf{v}_t$ is the observed token evidence, and $\mathbf{\Lambda}_t^{\mathrm{v}}$ is its reliability (precision). The query $\mathbf{q}_t$ instead specifies what component of the inferred latent state is exposed as the output.

## 4.2. Parallel Inference via Möbius and Affine Scans

A major obstacle to using Bayesian filters as sequence mixers is that classical filtering is *recursive* and thus appears inherently sequential. Our key observation is that, in *information form*, the updates admit a *compositional* structure that can be implemented with parallel scans.

**Information form.** We represent the Gaussian posterior by its precision and natural (canonical) parameter:

$$p(\mathbf{z}_t \mid \mathbf{v}_{1:t}) = \mathcal{N}(\boldsymbol{\mu}_t, \boldsymbol{\lambda}_t^{-1}), \qquad \boldsymbol{\eta}_t := \boldsymbol{\lambda}_t \odot \boldsymbol{\mu}_t. \quad (14)$$

This parameterisation is convenient because, in linear-Gaussian models, incorporating token evidence corresponds to *adding* canonical parameters (precisions and information means), while the remaining "predict" transformation induced by the dynamics takes a structured form. As we show next, these structured updates *compose associatively* across time, which is exactly what enables parallel prefix scans.

**Theorem 1** (Precision Update as a Möbius Transformation). *Let $\boldsymbol{\lambda}_t$ be the posterior precision at time $t$ in the diagonal linear-Gaussian model Equation (7)-Equation (9). Define $\boldsymbol{\phi}_t := \mathbf{k}_t^2 \odot \boldsymbol{\Lambda}_t^{\mathrm{v}}$. Then the map $\boldsymbol{\lambda}_{t-1} \mapsto \boldsymbol{\lambda}_t$ is a linear-fractional (Möbius) transform:*

$$\boldsymbol{\lambda}_t = \mathbf{M}_t(\boldsymbol{\lambda}_{t-1}) = \frac{\boldsymbol{\alpha}_t \odot \boldsymbol{\lambda}_{t-1} + \boldsymbol{\beta}_t}{\boldsymbol{\gamma}_t \odot \boldsymbol{\lambda}_{t-1} + \boldsymbol{\delta}_t}, \quad (15)$$

$$\mathbf{M}_t = \begin{pmatrix} \boldsymbol{\alpha}_t & \boldsymbol{\beta}_t \\ \boldsymbol{\gamma}_t & \boldsymbol{\delta}_t \end{pmatrix} = \begin{pmatrix} 1 + \bar{\mathbf{p}}_t \odot \boldsymbol{\phi}_t & \bar{\mathbf{a}}_t^2 \odot \boldsymbol{\phi}_t \\ \bar{\mathbf{p}}_t & \bar{\mathbf{a}}_t^2 \end{pmatrix}. \quad (16)$$

**Interpretation: precision track as uncertainty-driven gating.** While Equation (15) is nonlinear in $\boldsymbol{\lambda}_{t-1}$, it admits a simple "gate-like" rearrangement:

$$\boldsymbol{\lambda}_t = \underbrace{(\bar{\mathbf{a}}_t^2 + \bar{\mathbf{p}}_t \odot \boldsymbol{\lambda}_{t-1})^{-1} \odot \boldsymbol{\lambda}_{t-1}}_{\textit{(confidence history)}} + \underbrace{\boldsymbol{\phi}_t = \mathbf{k}_t^2 \odot \boldsymbol{\Lambda}_t^{\mathrm{v}}}_{\textit{(confidence in current token)}}.$$

$$\quad (17)$$

The shared denominator $\bar{\mathbf{a}}_t^2 + \bar{\mathbf{p}}_t \odot \boldsymbol{\lambda}_{t-1}$ introduces *history dependence*: as accumulated precision grows, the model naturally becomes more selective about incorporating new evidence. We name its inverse $\boldsymbol{\rho}_t := \mathbf{1} \oslash (\bar{\mathbf{a}}_t^2 + \bar{\mathbf{p}}_t \odot \boldsymbol{\lambda}_{t-1})$, so the history term of Equation (17) is $\boldsymbol{\rho}_t \odot \boldsymbol{\lambda}_{t-1}$. The *same* factor $\boldsymbol{\rho}_t$ reappears in the mean recursion below as the forget gate $\mathbf{f}_t = \boldsymbol{\rho}_t \odot \bar{\mathbf{a}}_t$ (Theorem 2), tying the precision and mean tracks together and inducing the input/forget-gate behaviour of the mean update (Equation (18)).[3]

**Corollary 1** (Precision Updates via Parallel Prefix Scan). *Given $\boldsymbol{\lambda}_0$ and matrices $\{\mathbf{M}_t\}_{t=1}^T$ from Theorem 1, the posterior precisions $\{\boldsymbol{\lambda}_t\}_{t=1}^T$ can be computed by a parallel prefix scan over $\{\mathbf{M}_t\}$ with $\mathcal{O}(T)$ work and $\mathcal{O}(\log T)$ depth.*

---

[3]In Equations (15) and (18) (and their state-expanded forms Equations (19) and (20)), the fraction bar $\mathbf{A}/\mathbf{B}$ and the inverse $(\cdot)^{-1}$ denote elementwise (Hadamard) division of equal-shape arrays, as in the $\mathbf{S}_t$ updates of Table 2.

**Theorem 2** (Mean Update as Affine Transformations). *Let $\boldsymbol{\eta}_t := \boldsymbol{\lambda}_t \odot \boldsymbol{\mu}_t$ be the posterior information mean and $\boldsymbol{\Lambda}_t^{\mathrm{v}}$ the value precision. Given the precision path $\{\boldsymbol{\lambda}_t\}$, the information mean evolves affinely:*

$$\boldsymbol{\eta}_t = \underbrace{(\bar{\mathbf{a}}_t^2 + \bar{\mathbf{p}}_t \odot \boldsymbol{\lambda}_{t-1})^{-1} \odot \bar{\mathbf{a}}_t}_{\mathbf{f}_t \;\textit{(history-dependent forget gate)}} \odot \boldsymbol{\eta}_{t-1} + \underbrace{\mathbf{k}_t \odot \boldsymbol{\Lambda}_t^{\mathrm{v}} \odot \mathbf{v}_t}_{\textit{token evidence}}.$$

$$\quad (18)$$

**Corollary 2** (Mean Updates via Parallel Prefix Scan). *The posterior information means $\{\boldsymbol{\eta}_t\}_{t=1}^T$ are computable via parallel prefix scan over the affine transformations in Theorem 2 in $\mathcal{O}(T)$ work and $\mathcal{O}(\log T)$ depth.*

**Special cases:** Under deterministic ($\mathbf{p}_t = 0$) and linear time-invariant (LTI) settings, the KLA updates reduce to convolutions computable in $\mathcal{O}(\log T)$ time via FFT (see Theorem 3 in the Appendix).

**Connection to online learning and test-time regression.** Recent sub-quadratic mixers can be unified as *online learners* (equivalently, *test-time regression*): each update is the closed-form minimiser of a per-token least-squares objective that balances staying near the decayed memory against absorbing the new token (Liu et al., 2025; Yang et al., 2023; 2024). KLA's posterior-mean update is exactly such a minimiser: it solves a *precision- (uncertainty-)weighted* least-squares objective, and the resulting moment-form recursion $\boldsymbol{\mu}_t = \boldsymbol{\eta}_t \oslash \boldsymbol{\lambda}_t$ resembles an *uncertainty-gated RNN* whose forget/write gates are precision ratios, supplied by the Möbius recursion of Theorem 1. Table 2 situates KLA among representative mixers;[4] the full template (objectives and all methods) is in Section E.

## 4.3. KLA as a Drop-in Probabilistic Sequence Mixer

To instantiate the above parallel Bayesian filter as a neural network layer, we parametrise the observation value and precision ($\mathbf{v}_t, \boldsymbol{\Lambda}_t^{\mathrm{v}}$) and the observation and readout operators ($\mathbf{k}_t, \mathbf{q}_t$) in terms of the layer input sequence $x_{1:T}$, with the observation variance parametrised to be positive. For the continuous stochastic prior (Section 4.1), we treat $\mathbf{a}_t$ and $\mathbf{p}_t$ as learnable, time-invariant parameters $\mathbf{a}$ and $\mathbf{p}$, unlike Mamba, where $\mathbf{a}_t$ is token-dependent and time-varying.

The block follows a fused MLP architecture (Figure 3a). Following Mamba, we pair the KLA sequence mixer with standard scaffolding: a 1D causal convolution (kernel size 4) with SiLU activation, residual connections, and simple nonlinearities; see Figure 5 and Appendix B for further details. We implement efficient Triton (Tillet et al., 2019) and CUDA kernels for the Linear and Möbius parallel scan primitives, but do not fuse the whole block into a single kernel, which other SSMs have shown to be necessary to minimise memory

---

[4]In the $\mathbf{S}_t$ updates, $\mathbf{A}/\mathbf{B}$ denotes elementwise (Hadamard) division of matrices.

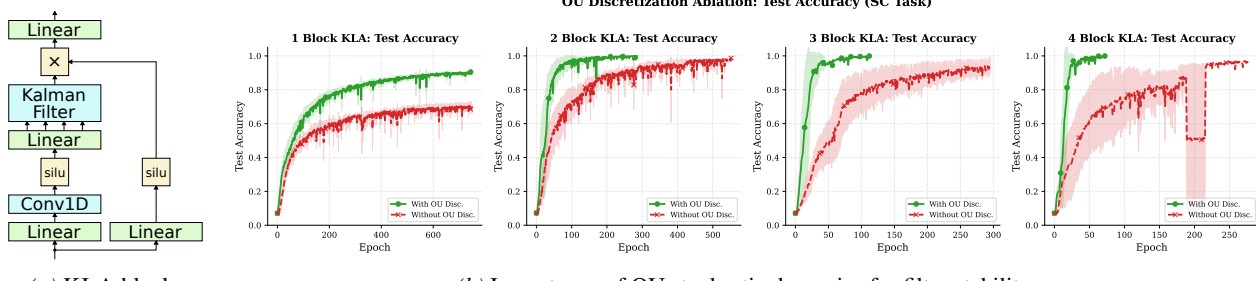

*(a)* KLA block.      *(b)* Importance of OU stochastic dynamics for filter stability.

*Figure 3.* **(a)** The KLA block follows Mamba's fused-MLP design, with the Kalman filter as a drop-in mixer (full architecture in Section B). **(b)** Ablating the OU prior dynamics and discretisation (Section 4.1): OU discretisation improves accuracy and learning stability, especially for deeper models.

| Method | Local online objective $\mathcal{L}_t(\mathbf{S})$ | State update | Gates from |
|---|---|---|---|
| Linear Attn. | $\|\mathbf{S} - \mathbf{S}_{t-1}\|_F^2 - 2\langle \mathbf{S}^\top \mathbf{k}_t, \mathbf{v}_t \rangle$ | $\mathbf{S}_t = \mathbf{S}_{t-1} + \mathbf{k}_t \mathbf{v}_t^\top$ | , |
| Mamba-1 (S6) | $\|\mathbf{S} - \bar{\mathbf{A}}_t \mathbf{S}_{t-1}\|_F^2 - 2\langle \mathbf{S}^\top \mathbf{k}_t, \mathbf{v}_t \rangle$ | $\mathbf{S}_t = \bar{\mathbf{A}}_t \mathbf{S}_{t-1} + \mathbf{k}_t \mathbf{v}_t^\top$ | $\bar{\mathbf{A}}_t$ |
| GDN | $\|\mathbf{S} - \alpha_t \mathbf{S}_{t-1}\|_F^2 - 2\langle \mathbf{S}^\top \mathbf{k}_t, \beta_t(\mathbf{v}_t - (\alpha_t \mathbf{S}_{t-1})^\top \mathbf{k}_t) \rangle$ | $\mathbf{S}_t = \alpha_t(\mathbf{I} - \beta_t \mathbf{k}_t \mathbf{k}_t^\top)\mathbf{S}_{t-1} + \beta_t \mathbf{k}_t \mathbf{v}_t^\top$ | $\alpha_t, \beta_t$ |
| **KLA (Ours)** | $\mathbf{\Lambda}_t^{\mathrm{prior}}\|\mathbf{S} - \bar{\mathbf{A}}\mathbf{S}_{t-1}\|_F^2 + \mathbf{\Lambda}_t^v \|\mathbf{S}^\top \mathbf{k}_t - \mathbf{v}_t\|^2$ | $\mathbf{S}_t = \bar{\mathbf{A}}\left(\mathbf{I} - \frac{\mathbf{k}_t^2(\mathbf{\Lambda}_t^v)^\top}{\mathbf{\Lambda}_t}\right)\mathbf{S}_{t-1} + \frac{\mathbf{k}_t(\mathbf{\Lambda}_t^v \mathbf{v}_t)^\top}{\mathbf{\Lambda}_t}$ | $\mathbf{\Lambda}_t^v$, Möbius $\mathbf{\Lambda}_t$ |

*Table 2.* Online-learning / test-time-regression view of sequence mixers (minimal). Each state update is the closed-form minimiser of the per-token objective $\mathcal{L}_t$ (for KLA, a *precision-weighted* least squares). Orange/green mark the forget/write gates; blue marks the precision weights and Möbius recursion supplying $\mathbf{\Lambda}_t$, unique to KLA. The full method set (DeltaNet, KDA, GDN-2) is in Table 6 in Section E.

movement and achieve peak performance, especially for the backward pass. The KLA posterior mean recurrence can also be unrolled into an equivalent matrix multiplication with a lower-triangular "attention" matrix, visualised in Figure 10 (Appendix G).

### 4.4. Training Loss

We use two training losses. **(i) Posterior mean:** logits $\ell_t = g_\theta(\mathbf{y}_t)$ are read out from the posterior mean $\mathbf{y}_t = \mathbf{q}_t \odot \boldsymbol{\mu}_t$ and trained with cross-entropy. **(ii) Log marginal likelihood:** we minimise the negative log marginal likelihood of the output token under the latent posterior (a Monte-Carlo objective), which provides an additional training signal for the posterior variances; the equations and details are given in Appendix B.

## 5. Experiments

We evaluate KLA as a *drop-in sequence mixer* for language modelling. Although KLA is derived from continuous-time Gaussian filtering, our goal is pragmatic: to test whether a Bayesian filtering view of selection/attention can retain the efficiency and accuracy of modern deterministic mixers, while yielding interpretable *uncertainty-driven gating* behaviour, and, crucially, whether such a probabilistic primitive can be *stacked* into deep networks and trained at scale.

**Experimental questions.** We address four questions:

- **(Q1) Compute and scaling:** how does the parallel implementation of KLA scale with sequence length, and how does it compare to a recurrent (time-stepped) Kalman

implementation in wall-clock and memory?

- **(Q2) Model quality as a single-block primitive:** as a drop-in mixer block, does KLA match or improve upon deterministic sub-quadratic alternatives (SSMs / linear attention) on controlled synthetic language-modelling tasks that probe core LM skills (long-range dependence, retrieval, and compression)?

- **(Q3) State-tracking expressivity:** does the fractional-linear structure induced by information-form filtering translate into stronger state-tracking capabilities (e.g., permutation composition) than linear SSM/attention baselines, while retaining scan-parallelisability?

- **(Q4) Stacking and scale:** can KLA be stacked into deep architectures and pretrained at the billion-token scale, matching attention and SSM/GLA backbones as a *standalone* mixer and *complementing* attention as a hybrid on zero-shot commonsense benchmarks?

Finally, we ablate architectural choices within the KLA block to isolate what enables stable deep stacking of a probabilistic primitive.

### 5.1. Baselines

We compare KLA against softmax attention and four state-of-the-art sub-quadratic mixers: **GPT** (Vaswani et al., 2017) (softmax attention, $\mathcal{O}(T^2)$ reference); **mLSTM** (Beck et al., 2024) (a modern LSTM with multiplicative interactions and exponential gating); **Mamba** (Gu & Dao, 2023) (a selective SSM with input-dependent transitions); **GDN** (Gated DeltaNet) (Yang et al., 2024), a delta-rule SSM that is the

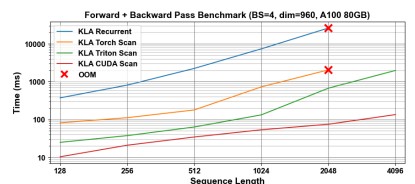

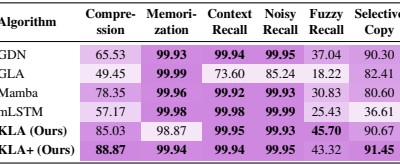

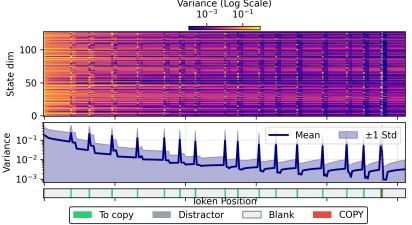

| Algorithm | Compre-ssion | Memori-zation | Context Recall | Noisy Recall | Fuzzy Recall | Selective Copy |
|---|---|---|---|---|---|---|
| GDN | 65.53 | **99.93** | **99.94** | **99.95** | 37.04 | 90.30 |
| GLA | 49.45 | **99.99** | 73.60 | 85.24 | 18.22 | 82.41 |
| Mamba | 78.35 | **99.96** | **99.92** | **99.93** | 30.83 | 80.60 |
| mLSTM | 57.17 | **99.98** | **99.98** | **99.99** | 25.43 | 36.61 |
| **KLA (Ours)** | 85.03 | 98.87 | **99.95** | **99.93** | **45.70** | 90.67 |
| **KLA+ (Ours)** | 88.87 | **99.94** | **99.94** | **99.95** | 43.32 | **91.45** |

*(a)* Training (fwd+bwd) runtime.     *(b)* MAD-Lab accuracy (%).     *(c)* Posterior variance.

*Figure 4.* **KLA as a single-block primitive: efficiency, quality, and uncertainty. (a)** Forward+backward (training) runtime vs. sequence length; scan-based KLA matches the $\mathcal{O}(\log T)$ profile of modern mixers (forward-only curve in Section G.3). **(b)** MAD-Lab synthetic-LM accuracy (GDN, GLA, Mamba, mLSTM, KLA, and probabilistic decoding KLA +); all methods are parameter- and state-matched; darker shading = higher accuracy. **(c)** Posterior variance on Selective Copy: spikes align with copy-relevant tokens, an interpretable filtering signal.

current state of the art among SSM LMs; and **GLA** (Gated Linear Attention) (Yang et al., 2023), a linear-attention variant unifying several linear RNN/SSM formulations. Models are matched on parameter count and effective state size; full configurations are in Appendix I.

## 5.2. Compute Scaling of Parallel KLA

Scaling probabilistic primitives to long contexts while retaining efficient training parallelism remains comparatively underexplored. To make the parallel formulation concrete, we benchmark three KLA implementations that progressively incorporate (i) a naive recurrent (time-stepped) Kalman update, (ii) a mathematically parallel formulation via a Torch associative scan, and (iii) hardware-oriented custom Triton and CUDA associative-scan kernels. The scan-based implementations scale efficiently with sequence length compared to the recurrent counterpart, confirming that *information-form filtering admits the same scan-parallel $\mathcal{O}(\log T)$ profile as modern SSM/GLA-style mixers*. The forward+backward (training) runtime curve is shown in Figure 4a; the forward-only (prompt-processing) curve and full setup are in Appendix G.3.

## 5.3. KLA as a Single-Block Primitive on Synthetic Language Modelling

**MAD synthetic LM suite.** We evaluate KLA and the baselines as a single-block primitive on the MAD-Lab benchmark (Poli et al., 2024), six synthetic token-manipulation tasks probing different LM skills (Table 7), matching parameter counts and effective state sizes under a common encoder-decoder protocol (Appendices H and I). Evaluating as a single block isolates the *update mechanism itself*, free of depth or memory-budget confounds.

**Results.** Figure 4b shows that KLA performs competitively across all tasks, matching or surpassing strong deterministic baselines. Gains are most pronounced on tasks that require *selection under corruption* (e.g., *Noisy Recall*, *Fuzzy Recall*, and *Selective Copy*), consistent with the interpretation of

KLA as performing uncertainty-weighted filtering. In addition, probabilistic decoding (KLA+, marginalisation under the learned posterior) yields further improvements on noise-corrupted tasks, suggesting that learning posterior variances can provide a useful training signal even without an explicit calibration objective.

**Posterior uncertainty visualisations.** To characterise how KLA filters information, we visualise the posterior variance on Selective Copy (Figure 4c). Rather than claiming calibrated uncertainty, we use it as a diagnostic of the model's filtering dynamics: variance generally decreases as evidence accumulates, while sharp spikes align with copy-relevant positions, marking timesteps the model treats as especially informative.

Additional experiments on long-context associative recall (MQAR) (Arora et al., 2023), where KLA outperforms Mamba and GLA, are reported in Appendix G.2.

## 5.4. State Tracking and Permutation-Group Expressivity

**Why $A_5$ matters.** A central open question is whether sub-quadratic sequence models can *track state* beyond highly-parallel shortcut computation (Merrill & Sabharwal, 2023; Merrill et al., 2024). Recent theory shows that common linear/diagonal SSMs (e.g. S4, Mamba), despite their recurrent form, share the bounded-depth expressivity limits of transformers, they are simulable by highly-parallel circuit classes (informally $\mathrm{TC}^0$) and cannot solve inherently sequential state-tracking whose difficulty grows with sequence length. A canonical example is the word problem for the alternating group $A_5$ (the smallest non-solvable subgroup of $S_5$), which is $\mathrm{NC}^1$-complete and serves as a minimal benchmark for *hard state tracking* (Merrill et al., 2024).

**Result.** Following Merrill et al. (2024), Figure 1(a) shows that KLA solves the $A_5$ task at constant depth with only 1-2 layers, whereas linear SSM/attention baselines require depth that grows with sequence length in this regime. Among the parallel-trainable architectures we consider,

| Model | LAMB. | HellaS.[n] | PIQA | Arc-E | Arc-C[n] | WinoGr. | OBQA | BoolQ | Average |
|---|---|---|---|---|---|---|---|---|---|
| GPT-45M | **16.00** | 28.60 | 58.00 | 40.20 | 20.60 | 50.00 | **27.40** | 37.40 | 34.78 |
| Mamba-45M | 9.20 | **29.20** | **59.80** | 37.20 | **25.40** | 48.80 | 25.40 | 61.60 | 37.08 |
| GDN-45M | **16.00** | 28.20 | 58.40 | 39.00 | 23.40 | **52.60** | 27.20 | 45.20 | 36.25 |
| KLA-45M | 7.60 | 28.80 | 58.60 | 40.60 | 23.20 | 52.20 | **27.40** | **63.00** | 37.68 |
| **GPT+KLA-45M** | **16.00** | 28.20 | 58.60 | **42.80** | 23.00 | 50.80 | 24.40 | 62.80 | **38.33** |
| GPT-180M | **28.80** | 32.20 | 63.40 | 51.00 | 27.00 | 49.20 | 31.20 | 54.80 | 42.20 |
| Mamba-180M | 24.00 | 30.60 | 63.00 | 51.00 | 27.20 | **53.80** | **33.00** | 55.80 | 42.30 |
| GDN-180M | 27.80 | 31.60 | 60.40 | 45.60 | 25.80 | 53.40 | 30.00 | 60.80 | 41.93 |
| KLA-180M | 22.40 | 34.00 | 63.00 | 50.00 | 26.00 | 53.60 | 32.40 | 60.80 | 42.78 |
| **GPT+KLA-180M** | 30.20 | **34.40** | **64.20** | **55.80** | **29.00** | 51.80 | 31.00 | **63.20** | **44.95** |

*Table 3.* **Language modelling at two academic scales.** Zero-shot accuracy (%, ↑) on eight commonsense benchmarks at 45M (1.8B tokens) and 180M (10.9B tokens); [n] denotes length-normalised accuracy ($acc_n$). **Bold** = best within each size block, underline = second. Rows above the dashed line are standalone mixers; the row below it is the hybrid GPT+KLA, in which only the *final* attention layer of GPT is replaced with a KLA block.

KLA is the only one to solve the task at constant depth; concurrent work shows linear RNNs can also acquire state tracking through negative eigenvalues (Grazzi et al., 2025), whereas KLA achieves it through the nonlinearity of the Möbius precision recurrence. This is consistent with KLA's *fractional-linear* (information-form) updates: each step stays scan-parallelisable, yet the state evolution is richer than a linear recurrence, placing KLA beyond $TC^0$ while preserving the parallel-scan structure of efficient SSM/linear-attention models.

### 5.5. Stacking KLA for Language Modelling at Scale

The experiments above isolate KLA as a single (1-2 block) primitive. We now ask whether it can be *stacked into deep networks and trained at scale*: we pretrain KLA and all baselines from scratch on OpenFineWeb-Edu at **45M** (1.8B tokens) and **180M** (10.9B tokens) parameters, with matched optimiser and token budget, reporting zero-shot accuracy on eight commonsense benchmarks (full protocol in Appendix I). To our knowledge this is the first large-scale from-scratch pretraining of a stacked probabilistic-filtering primitive propagating belief states between blocks, and we regard stable training at this scale as a contribution in itself. Stacking also raises an interface question absent for deterministic mixers: KLA emits a full posterior at every step; as a first iteration we propagate only the *mean* between blocks (a single posterior sample performs comparably), leaving full-posterior propagation to future work.

**Standalone KLA is competitive.** Table 3 shows KLA is competitive with softmax attention, Mamba, and GDN at both scales (with per-task strengths on HellaSwag and PIQA). Its advantage is largest at shallow depth and narrows as baselines add layers (in our 24-layer experiments), as predicted by our expressivity analysis (Section 5.4). It underperforms on the *clean* copying task LAMBADA, consistent with uncertainty-weighted updates favouring robust over verbatim recall (cf. Section 5.3).

**A single KLA layer improves a GPT.** Unlike prior hybrids that interleave many layers, we replace only the *final* layer of a softmax-attention GPT with an SSM-like block (KLA, Mamba, or GDN), motivated by evidence that uncertainty concentrates in later layers (Agarwal et al., 2025). At 180M, GPT+KLA improves over pure GPT and over the analogous GPT+Mamba and GPT+GDN hybrids on average and most tasks (Table 3; Figure 1(b)), showing KLA is an effective drop-in *complement* to attention, not merely a substitute.

### 5.6. Ablations

Finally, we isolate the two modelling choices that matter most. Stable *deep stacking* of a probabilistic primitive was possible *only* because of the mean-reverting Ornstein-Uhlenbeck (OU) prior (Section 4.1): naive stacking was unstable and standard fixes (gradient clipping, lower learning rates) did not help, whereas the OU prior keeps the latent dynamics bounded, letting KLA stack to 12 and 24 blocks stably (Figure 3b confirms improved stability and accuracy on Selective Copy, especially at depth). A second ablation (Figure 7, Appendix G.1) shows the process-noise parameter supplies the nonlinear gating: setting $p_t=0$ collapses the Möbius precision recurrence to a fixed-gate linear recurrence and degrades MAD accuracy by 49.8 points on average.

## 6. Conclusion

We introduced the KLA layer, a scalable probabilistic sequence mixer that casts language modelling as state estimation, with tractable posterior inference via parallel scans and gating strictly more expressive than deterministic linear SSMs and RNNs. We show the Kalman filter admits an *exact* parallelisation in information form, and integrates into language modelling at scale, as a standalone mixer or attention hybrid. Our experiments show its promise for language modelling, state tracking, and probabilistic sequence modelling (**Limitations** and future work in Appendix A).

## Impact Statement

This paper presents work whose goal is to advance the field of Machine Learning. There are many potential societal consequences of our work, none of which we believe must be specifically highlighted here. The KLA layer provides principled uncertainty quantification, which could benefit applications requiring calibrated predictions, though, as with any language modelling primitive, deployment should consider potential misuse.

## Acknowledgements

We thank Henry Gouk, Nikolay Malkin, Thomas Lee, and Stephan Kostov for their valuable feedback on this draft at various stages. We also thank Wouter Boomsma, Frederikke Isa Marin, Jose Cano Reyes, and Jude Harris for the informative discussions over the course of this research. This project has received funding from the European Union's Horizon Europe research and innovation programme under grant agreement No. 101120726. This work was funded by UK Research and Innovation (UKRI) under the UK government's Horizon Europe funding Guarantee 10085198. The authors acknowledge the use of resources provided by the Isambard-AI National AI Research Resource (AIRR) (McIntosh-Smith et al., 2024). Isambard-AI is operated by the University of Bristol and is funded by the UK Government's Department for Science, Innovation and Technology (DSIT) via UK Research and Innovation; and the Science and Technology Facilities Council [ST/AIRR/I-A-I/1023].

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

# Appendix

**Appendix outline**

## A. Limitations and Future Work

This paper is a *first iteration* on a fundamental algorithmic family, Bayesian filters as parallelisable sequence-modelling primitives, and we deliberately start from the simplest possible assumptions: diagonal, linear time-invariant (LTI) dynamics with a linear-Gaussian observation model. Richer yet compute-friendly variants (e.g., structured or input-dependent dynamics, non-diagonal couplings, and nonlinear or learned observation models) are natural next steps that could close the gap to full Kalman filtering while retaining the scan-parallel profile.

We have also not explicitly exploited the posterior covariance for applications where uncertainty quantification may be critical (e.g., epistemic prompt uncertainty, hallucination detection, and out-of-distribution prompt detection), which are likely areas where KLA could excel; we leave this for future work.

Finally, because the underlying state-space formulation is inherently continuous, a promising direction is to move beyond the language domain and train vision and multimodal foundation models over continuous modalities, where explicit belief-state uncertainty may be especially natural.

## B. Architecture Details

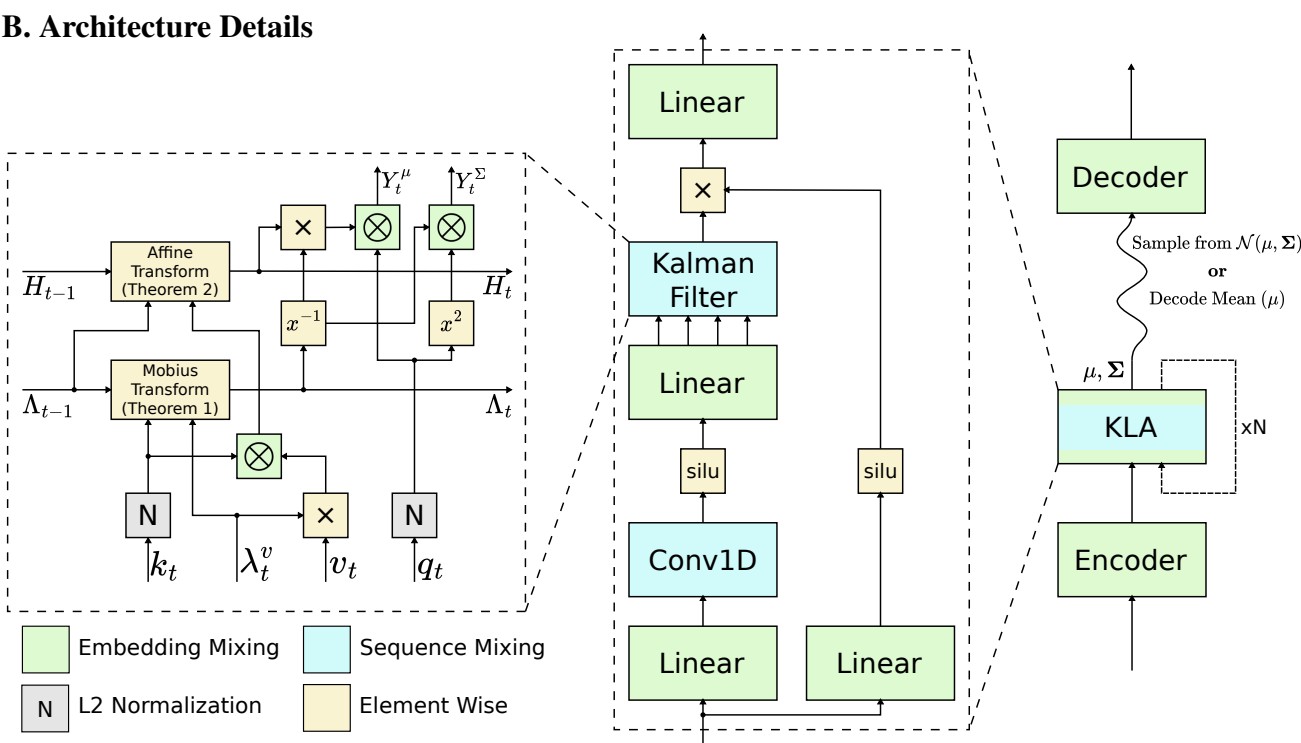

*Figure 5.* **Model architecture used in our experiments.** The KLA block is a drop-in replacement for Transformer or SSM blocks. We follow a similar block structure to Mamba, fusing the sequence mixing and gated MLP into a single block. We additionally employ QK-Norm and use an expansion factor of 1. Tokens are embedded and passed through a *sequence mixer*, then decoded to logits. The sequence mixer is a drop-in primitive that can be instantiated as attention variants, or other SSM/Filter modules. Many mixers are paired with standard *scaffolding*-e.g., 1D convolutions, residual/skip connections, and simple nonlinearities (elementwise multiplication or activations)-shown on the right.

**State-expanded matrix form of the scans.** The `Mobius-Scan` and `Linear-Scan` steps are the state-expanded ($N>1$) matrix versions of the per-channel recursions Equations (15) and (18). Adopting the column-vector convention of Table 2 ($\mathbf{k}_t \in \mathbb{R}^N$, $\mathbf{v}_t, \mathbf{\Lambda}_t^{\mathrm{v}} \in \mathbb{R}^D$, information-mean state $\mathbf{H}_t = \boldsymbol{\eta}_t$ and precision $\mathbf{\Lambda}_t$ in $\mathbb{R}^{N \times D}$; see footnote 2), they read

$$\mathbf{\Lambda}_t = \frac{(\mathbf{1} + \bar{\mathbf{P}} \odot \mathbf{\Phi}_t) \odot \mathbf{\Lambda}_{t-1} + \bar{\mathbf{A}}^2 \odot \mathbf{\Phi}_t}{\bar{\mathbf{P}} \odot \mathbf{\Lambda}_{t-1} + \bar{\mathbf{A}}^2}, \qquad \mathbf{\Phi}_t = \mathbf{k}_t^2 \left(\mathbf{\Lambda}_t^{\mathrm{v}}\right)^\top, \tag{19}$$

$$\mathbf{H}_t = \mathbf{F}_t \odot \mathbf{H}_{t-1} + \mathbf{k}_t \left(\mathbf{\Lambda}_t^{\mathrm{v}} \odot \mathbf{v}_t\right)^\top, \qquad \mathbf{F}_t = \bar{\mathbf{A}} \oslash \left(\bar{\mathbf{A}}^2 + \bar{\mathbf{P}} \odot \mathbf{\Lambda}_{t-1}\right), \tag{20}$$

where $\odot, \oslash$ act elementwise on $N \times D$ arrays, and the rank-one outer products $\mathbf{k}_t^2(\mathbf{\Lambda}_t^{\mathrm{v}})^\top$ and $\mathbf{k}_t(\mathbf{\Lambda}_t^{\mathrm{v}} \odot \mathbf{v}_t)^\top$ supply the per-slot evidence (mirroring the $\mathbf{k}_t \mathbf{v}_t^\top$ term of Table 2). Here $\mathbf{H}_t = \boldsymbol{\eta}_t$ is the *information* (natural-parameter) mean; the

---

**Algorithm 1** Kalman Linear Attention

---

1: **Input: X** : (B, T, 1, D)
2: **Output $\mathbf{Y}^\mu$** : (B, T, 1, D)
3: **Output (Optional) $\mathbf{Y}^\Sigma$** : (B, T, 1, D)
4: $\mathbf{A}, \mathbf{P}, \boldsymbol{\Delta}$ : (N, D) ← Parameters                                             ▷ A: Gating/decay, P: Process noise, $\Delta$: Discretisation step size
5: $\mathbf{K}$ : (B, T, N, 1) ← $f_{\theta_k}(\mathbf{X})$                                             ▷ Observation operator
6: $\mathbf{Q}$ : (B, T, 1, N) ← $f_{\theta_q}(\mathbf{X})$                                             ▷ Readout operator
7: $\mathbf{V}, \boldsymbol{\Lambda}^{\mathrm{v}}$ : (B, T, 1, D) ← $f_{\theta_{v,\lambda^{\mathrm{v}}}}(\mathbf{X})$                              ▷ Observation mean and precision, $\lambda^{\mathrm{v}} \in \mathbb{R}^+$
8: $\bar{\mathbf{A}}$ : (N, D) ← OU-Discretisation($\mathbf{A}, \boldsymbol{\Delta}$)                              ▷ See Equation (8)
9: $\bar{\mathbf{P}}$ : (N, D) ← OU-Discretisation($\mathbf{A}, \mathbf{P}, \boldsymbol{\Delta}$)
10: $\boldsymbol{\Lambda}$ : (B, T, N, D) ← Mobius-Scan($\mathbf{K}, \boldsymbol{\Lambda}^{\mathrm{v}}, \bar{\mathbf{P}}, \bar{\mathbf{A}}$)                              ▷ matrix form: Equation (19)
11: $\mathbf{H}$ : (B, T, N, D) ← Linear-Scan($\mathbf{K}, \boldsymbol{\Lambda}^{\mathrm{v}}, \mathbf{V}, \boldsymbol{\Lambda}, \bar{\mathbf{P}}, \bar{\mathbf{A}}$)                ▷ information mean $\eta$; matrix form: Equation (20)
12: $\mathbf{S}$ : (B, T, N, D) ← $\mathbf{H} \oslash \boldsymbol{\Lambda}$                              ▷ moment-form state ($\mathbf{S}_t$ of Table 2): $\boldsymbol{\mu} = \boldsymbol{\eta} \oslash \boldsymbol{\lambda}$
13: $\mathbf{Y}^\mu$ : (B, T, 1, D) ← $\mathbf{QS}$
14: **if** Decode Variance **then**
15:     $\mathbf{Y}^\Sigma$ : (B, T, 1, D) ← $\mathbf{Q}^2 \boldsymbol{\Lambda}^{-1}$
16:     **Return: $\mathbf{Y}^\mu, \mathbf{Y}^\Sigma$**
17: **else**
18:     **Return: $\mathbf{Y}^\mu$**
19: **end if**

---

*moment-form* state of Table 2 is $\mathbf{S}_t = \mathbf{H}_t \oslash \boldsymbol{\Lambda}_t$ (i.e. $\boldsymbol{\mu}_t = \boldsymbol{\eta}_t \oslash \boldsymbol{\lambda}_t$), and the posterior mean reads out as $\mathbf{y}_t = \mathbf{q}_t \mathbf{S}_t$. Setting $N{=}1$ recovers the scalar recursions Equations (15) and (18).

**Architecture.** We use a consistent language modelling architecture across all models and baselines, as illustrated in Figure 5. The architecture follows a standard pattern: input tokens are embedded, passed through a sequence mixer layer, and then decoded to output logits. The sequence mixer serves as a drop-in primitive that can be instantiated as KLA, Mamba, GLA, or other SSM/attention variants.

Following the Mamba design, we employ standard scaffolding components paired with the sequence mixer: a 1D causal convolution with kernel size 4 and SiLU activation, along with residual connections and simple nonlinearities (elementwise multiplication). We use standard state expansion (see Section 3) and is set to 16 (N or sometimes called d_state in SSM codebases) as default similar to (Gu & Dao, 2023). More extensive exploration of scaffolding configurations optimally suited to the KLA filtering mixer can be a direction for future work.

We implemented KLA with PyTorch (Paszke et al., 2019) and used the Adam optimiser (Kingma & Ba, 2015) for training the models.

**Training loss.** We consider two training losses. Throughout, the decoder is applied to the query-conditioned readout $\mathbf{y}_t = \mathbf{q}_t \odot \mathbf{z}_t$; when the readout noise is zero this is a deterministic linear map, so conditioning on $\mathbf{z}_t$ or $\mathbf{y}_t$ is equivalent up to this transformation.

**(i) Posterior mean:** logits $\ell_t = g_\theta(\mathbf{y}_t)$ are obtained from the posterior-mean readout $\mathbf{y}_t = \mathbf{q}_t \odot \boldsymbol{\mu}_t$ and trained with cross-entropy.

**(ii) Log marginal likelihood:** minimises the negative log marginal likelihood of the output token $o_t$ under the latent posterior,

$$-\log p_\theta(o_t \mid \mathbf{X}) = -\log \int \underbrace{p_\theta(o_t \mid \mathbf{y}_t)}_{\text{decoder}} \underbrace{p_\theta(\mathbf{z}_t \mid \mathbf{X})}_{\text{KLA posterior}} d\mathbf{z}_t$$

$$\approx -\log\left(\frac{1}{S}\sum_{s=1}^{S} p_\theta(o_t \mid \mathbf{y}_t^{(s)})\right), \tag{21}$$

with $\mathbf{z}_t^{(s)} \sim p_\theta(\mathbf{z}_t \mid \mathbf{X})$ and $\mathbf{y}_t^{(s)} = \mathbf{q}_t \odot \mathbf{z}_t^{(s)}$. Equivalently, the estimator can be written as a $\mathrm{logsumexp}$:

$$
\begin{aligned}
&- \log p_\theta(o_t \mid \mathbf{X}) \\
&\approx -\log\left( \frac{1}{S} \sum_{s=1}^{S} p_\theta(o_t \mid \mathbf{y}_t^{(s)}) \right) \\
&= -\underset{s=1}{\overset{S}{\mathrm{logsumexp}}}\left( \log p_\theta(o_t \mid \mathbf{y}_t^{(s)}) \right) \; + \; \log S,
\end{aligned}
\tag{22}
$$

which reduces to cross-entropy when $S=1$. We show this for a single time-step; it can be trivially extended across timesteps.

| Attention | Deterministic SSM / GLA | KLA (probabilistic) |
|---|---|---|
| Query operator $\mathbf{q}_t$ | Output/readout map $\mathbf{C}_t$ | Query/readout operator $\mathbf{q}_t$ |
| Key operator $\mathbf{k}_t$ | Input to latent map $\mathbf{b}_t$ | Observation model/operator $\mathbf{k}_t$ |
| Value $\mathbf{v}_t$ | Token input/control $\mathbf{u}_t$ | Noisy observation $\mathbf{v}_t$ |
| Output $\mathbf{y}_t$ | Output $\mathbf{y}_t$ | Output $\mathbf{y}_t$ |
| *Hidden state (implicit in attention, explicit in SSMs/*KLA*):* | | |
| - | Hidden state $\mathbf{h}_t$ | Posterior belief $\mathcal{N}(\boldsymbol{\mu}_t, \boldsymbol{\Lambda}_t^{-1})$ over $\mathbf{z}_t$ |
| - | State decay $\bar{\mathbf{a}}_t = e^{-\mathbf{a}(t)\Delta t}$ | State decay $\bar{\mathbf{a}}_t = e^{-\mathbf{a}\Delta t}$ |
| *Uncertainty terms (explicit in* KLA *only):* | | |
| - | - | Value precision $\boldsymbol{\Lambda}_t^{\mathrm{v}}$ |
| - | - | State precision $\boldsymbol{\lambda}_t$ |
| - | - | Process noise $\bar{\mathbf{p}}_t = \frac{\mathbf{p}^2}{2\mathbf{a}} \odot \left(1 - e^{-2\mathbf{a}\Delta t}\right)$ |

*Table 4.* Notation alignment across attention, SSMs, and KLA. Precisions ($\boldsymbol{\lambda}, \boldsymbol{\Lambda}^{\mathrm{v}}, \boldsymbol{\Lambda}^{\mathrm{p}}$) are inverse variances encoding confidence: higher precision means lower uncertainty.

## C. Notation

Throughout the paper we use the following convention:

| Symbol | Meaning |
|---|---|
| $t$ | Time index |
| $z_t$ | Latent state at time $t$ |
| $v_t$ | Value (observed token/features) at time $t$ |
| $A_t$ | State transition matrix |
| $P_t$ | Process noise covariance (discretized: $\bar{P}_t$) |
| $C_t$ | Observation (likelihood) matrix |
| $\Lambda_t^{\mathrm{v}}$ | Value precision (inverse observation variance) |
| $\mathcal{N}(\mu, \Sigma)$ | Gaussian in *moment* form |
| $\mathcal{N}(\eta, \Lambda)$ | Gaussian in *canonical* form; $\eta = \Lambda\mu, \; \Lambda = \Sigma^{-1}$ |
| $\mu_t^{\mathrm{prior}}, \Sigma_t^{\mathrm{prior}}$ | Predictive prior mean/cov at $t$ (canonical: $\eta_t^{\mathrm{prior}}, \Lambda_t^{\mathrm{prior}}$) |
| $\mu_t, \Sigma_t$ | Posterior mean/cov at $t$ (canonical: $\eta_t, \Lambda_t$) |
| **Beliefs (linear-Gaussian case):** | |
| $z_t \mid v_{1:t-1} \sim \mathcal{N}(\mu_t^{\mathrm{prior}}, \Sigma_t^{\mathrm{prior}})$   (predictive prior) | |
| $z_t \mid v_{1:t} \sim \mathcal{N}(\mu_t, \Sigma_t)$   (posterior) | |

**Diagonal (per-dimension) specialisation.** When matrices are diagonal, we identify them with their diagonal vectors and switch to bold lowercase: $A_t \equiv \mathbf{a}_t, B_t \equiv \mathbf{b}_t, K_t \equiv \mathbf{k}_t$ (key), $P_t \equiv \mathbf{p}_t$ (process noise); value precision $\Lambda_t^{\mathrm{v}}$ is treated as a scalar. All algebra becomes *elementwise*: products use the Hadamard operator $\odot$, superscripts like $\mathbf{a}_t^2$ denote elementwise squares, and $(\cdot)^{-1}$ on lowercase vectors means elementwise inverse.

| Full (matrix) | Diagonal notation (this paper) | Implication (elementwise) |
|---|---|---|
| $A_t$ | $\mathbf{a}_t$ | $(A_t \mathbf{x}) = \mathbf{a}_t \odot \mathbf{x}; \quad A_t^2 \Rightarrow \mathbf{a}_t^2$ |
| $K_t$ | $\mathbf{k}_t$ | $K_t \Sigma K_t^\top \Rightarrow \mathbf{k}_t^2 \odot \Sigma$ (key) |
| $P_t$ | $\mathbf{p}_t$ | Process noise cov.; $\bar{P}_t \Rightarrow \bar{\mathbf{p}}_t$ (discretized) |
| $\Lambda_t^{\mathrm{v}}$ | $\Lambda_t^{\mathrm{v}}$ | Scalar value precision (variance $= (\Lambda_t^{\mathrm{v}})^{-1}$) |

# D. Extended Background

### D.1. Möbius / Fractional-Linear Transformations.

A Möbius (a.k.a. fractional-linear) transformation is the map

$$z \;\mapsto\; \frac{az + b}{cz + d}, \qquad ad - bc \neq 0.$$

It generalises the familiar affine form ($c = 0$ gives $az + b$) by introducing an additional denominator term. Each transform is represented (up to multiplication by a nonzero scalar) by a $2 \times 2$ matrix $M = \begin{pmatrix} a & b \\ c & d \end{pmatrix}$ and acts on a scalar $z$ via the linear-fractional rule

$$M(z) \;\coloneqq\; \frac{az + b}{cz + d}.$$

Composing multiple transforms amounts to multiplying their representing matrices, so associativity and invertibility (whenever $\det M \neq 0$) follow directly. Geometrically, Möbius transforms can be viewed as compositions of translations, rotations/scalings, and a crucial nonlinear ingredient-*inversion*-which affine transformations cannot capture. The denominator also provides a built-in *self-normalisation*: as the state grows, it is automatically rescaled, producing saturation and stability effects reminiscent of gating or normalisation layers, but achieved directly through the update itself.

### D.2. Information Form of Gaussian Distributions and Information Filters.

Gaussians belong to the exponential family (Nielsen & Garcia, 2009), so the posterior can be written in *canonical* (information) form with precision $\Lambda_t = \Sigma_t^{-1}$ and natural parameter $\eta_t = \Lambda_t \mu_t$. Filtering recursions in this parameterisation are known as *information filters (IF)* (Anderson & Moore, 2005; Khan, 2005). They are algebraically equivalent to the standard (moment) Kalman updates; only the representation of the Gaussian belief differs. Crucially, the canonical posterior updates are *linear-additive* in $(\eta, \Lambda)$, a property we leverage for scalable sequence modelling in Section 4.

### D.3. Control vs Observation View Of Token Processing

State-space models can be understood from two complementary perspectives (Figure 6):

**Observation view (traditional SSM/Kalman filter).** In classical Bayesian filtering, input tokens are treated as *noisy observations* of a hidden state. The model assumes: (i) a latent state $z_t$ evolves over time via learned dynamics, and (ii) each token $v_t$ is a noisy measurement generated from this state. The goal is to *infer* the hidden state from the observed tokens-asking "what underlying state could have produced these observations?"

**Control view (Mamba/S6).** In contrast, Mamba treats input tokens as *control signals* that directly drive the state. There is no observation noise or inference step; tokens deterministically update the hidden state via $z_t = A_t z_{t-1} + B_t u_t$. The state simply accumulates information from inputs-asking "how should I update my memory given this new input?"

**Why this matters.** The observation view naturally provides uncertainty quantification: since tokens are noisy measurements, the model maintains a posterior *distribution* over states rather than a point estimate. This is the foundation of KLA-we treat language modelling as Bayesian filtering where tokens inform our belief about an underlying semantic state.

### D.4. Kalman and Information Filters

D.4.1. BAYESIAN FILTERING AS POSTERIOR INFERENCE.

Bayesian filters (Kalman, information-filter variants) are *inference schemes*-procedures that compute posterior beliefs about latent states from noisy data. The Kalman filter assumes a Gaussian linear state-space model: the latent evolves via a (possibly time-varying) linear transition $A_t$ with unmodelled variability captured by process noise $Q_t$, and each observed token feature $o_t$ is generated by an observation model $C_t$ with observation noise covariance $\Sigma_t^{\text{obs}}$. The filter then computes $p(z_t \mid o_{1:t})$ sequentially by combining the dynamics prior with token evidence.

In this linear-Gaussian setting, inference has closed-form solutions: in the *moment parametrisation* $(\mu_t, \Sigma_t)$ the scheme is called the **Kalman filter**, while in the *canonical parametrisation* $(\eta_t, \Lambda_t)$ it is called the **information filter**. Both are algebraically equivalent, differing only in how Gaussian beliefs are represented.

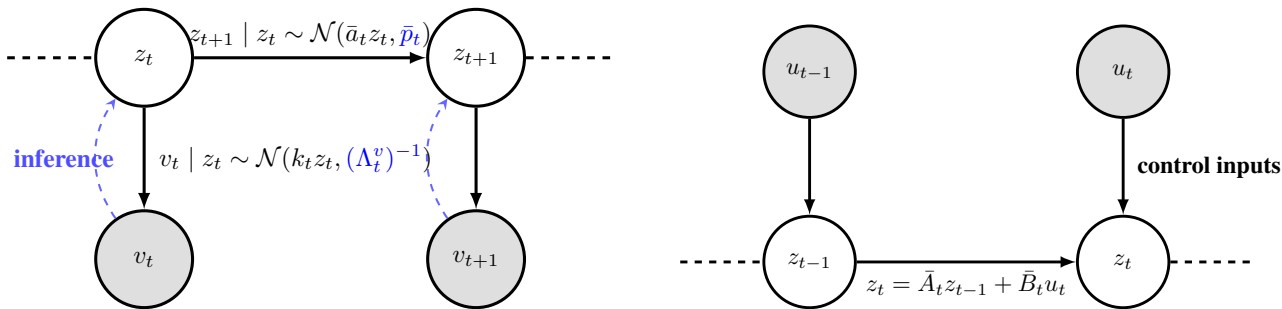

*(a)* Observation view: tokens $v_t$ are noisy observations of latent state $z_t$. Blue arrows show inference direction.

*(b)* Control view: tokens $u_t$ directly drive state updates. No inference needed.

*Figure 6.* Two views of state-space sequence models. **Left:** Bayesian/observation view-tokens are measurements, inference recovers hidden state. **Right:** Control view (Mamba)-tokens are inputs that deterministically update state.

| Belief / Parameters | Kalman Filter (moment form) | Information Filter (canonical form) |
|---|---|---|
| **Prior / Prediction** $z_t \mid v_{1:t-1} \sim \mathcal{N}(\mu_t^{\text{prior}}, \Sigma_t^{\text{prior}})$ | $\mu_t^{\text{prior}} = A_t \mu_{t-1}$ $\Sigma_t^{\text{prior}} = A_t \Sigma_{t-1} A_t^\top + P_t$ | $\eta_t^{\text{prior}} = \Lambda_t^{\text{prior}} A_t \Lambda_{t-1}^{-1} \eta_{t-1}$ $\Lambda_t^{\text{prior}} = \left( A_t \Lambda_{t-1}^{-1} A_t^\top + P_t \right)^{-1}$ |
| **Posterior / Update** $z_t \mid v_{1:t} \sim \mathcal{N}(\mu_t, \Sigma_t)$ | $\mu_t = \mu_t^{\text{prior}} + K_t \left( v_t - C_t \mu_t^{\text{prior}} \right)$ $\Sigma_t = (I - K_t C_t) \Sigma_t^{\text{prior}}$ $K_t = \Sigma_t^{\text{prior}} C_t^\top (C_t \Sigma_t^{\text{prior}} C_t^\top +$ $(\Lambda_t^{\text{v}})^{-1})^{-1}$ | $\Lambda_t = \Lambda_t^{\text{prior}} + C_t^\top \Lambda_t^{\text{v}} C_t$ $\eta_t = \eta_t^{\text{prior}} + C_t^\top \Lambda_t^{\text{v}} v_t$ |

*Table 5.* Two equivalent views at time $t$: moment (KF) vs. canonical (IF). Superscript "prior" denotes the predictive prior; unadorned symbols denote the posterior.

**Intuition on noise.**  In our linear-Gaussian state-space model, the "noise" terms $w_t$ and $v_t$ are not claims that language or the world is intrinsically random; they compactly represent *uncertainty* and unmodeled effects (domain shift, annotation/embedding error, missing context). Their covariances $P_t$ and $\Sigma_t^{\text{obs}}$ calibrate trust in dynamics vs. data-larger $P_t$ yields more adaptivity (faster forgetting), while larger $\Sigma_t^{\text{obs}}$ leans on the prior-producing principled "gates" via precision ratios. In contrast, 'noise' in diffusion models is an *algorithmic corruption* deliberately added in a forward process to define a denoising/score-matching objective; its schedule ($\beta_t$) is chosen for optimisation/SNR shaping rather than for model measurement uncertainty.

| Method | Local online objective $\mathcal{L}_t(\mathbf{S})$ | State update* | Gates from |
|---|---|---|---|
| *Correlation write into a (decayed) state* | | | |
| Linear Attn. | $\|\mathbf{S} - \mathbf{S}_{t-1}\|_F^2 - 2\langle\mathbf{S}^\top\mathbf{k}_t, \mathbf{v}_t\rangle$ | $\mathbf{S}_t = \mathbf{S}_{t-1} + \mathbf{k}_t\mathbf{v}_t^\top$ | - |
| Mamba-1 (S6) | $\|\mathbf{S} - \bar{\mathbf{A}}_t\mathbf{S}_{t-1}\|_F^2 - 2\langle\mathbf{S}^\top\mathbf{k}_t, \mathbf{v}_t\rangle$ | $\mathbf{S}_t = \bar{\mathbf{A}}_t\mathbf{S}_{t-1} + \mathbf{k}_t\mathbf{v}_t^\top$ | $\mathbf{A}, \bar{\mathbf{A}}_t$ |
| Mamba-2 | $\|\mathbf{S} - \alpha_t\mathbf{S}_{t-1}\|_F^2 - 2\langle\mathbf{S}^\top\mathbf{k}_t, \mathbf{v}_t\rangle$ | $\mathbf{S}_t = \alpha_t\mathbf{S}_{t-1} + \mathbf{k}_t\mathbf{v}_t^\top$ | $a, \alpha_t$ |
| *Delta (residual) write* | | | |
| DeltaNet | $\|\mathbf{S} - \mathbf{S}_{t-1}\|_F^2 - 2\langle\mathbf{S}^\top\mathbf{k}_t, \beta_t(\mathbf{v}_t - \mathbf{S}_{t-1}^\top\mathbf{k}_t)\rangle$ | $\mathbf{S}_t = (\mathbf{I} - \beta_t\mathbf{k}_t\mathbf{k}_t^\top)\mathbf{S}_{t-1} + \beta_t\mathbf{k}_t\mathbf{v}_t^\top$ | $\beta_t$ |
| Gated DeltaNet | $\|\mathbf{S} - \alpha_t\mathbf{S}_{t-1}\|_F^2 - 2\langle\mathbf{S}^\top\mathbf{k}_t, \beta_t(\mathbf{v}_t - (\alpha_t\mathbf{S}_{t-1})^\top\mathbf{k}_t)\rangle$ | $\mathbf{S}_t = \alpha_t(\mathbf{I} - \beta_t\mathbf{k}_t\mathbf{k}_t^\top)\mathbf{S}_{t-1} + \beta_t\mathbf{k}_t\mathbf{v}_t^\top$ | $\alpha_t, \beta_t$ |
| KDA (Team et al., 2025) | $\|\mathbf{S} - \mathbf{D}_t\mathbf{S}_{t-1}\|_F^2 - 2\langle\mathbf{S}^\top\mathbf{k}_t, \beta_t(\mathbf{v}_t - (\mathbf{D}_t\mathbf{S}_{t-1})^\top\mathbf{k}_t)\rangle$ | $\mathbf{S}_t = (\mathbf{I} - \beta_t\mathbf{k}_t\mathbf{k}_t^\top)\mathbf{D}_t\mathbf{S}_{t-1} + \beta_t\mathbf{k}_t\mathbf{v}_t^\top$ | $\mathbf{D}_t, \beta_t$ |
| GDN-2 (Hatamizadeh et al., 2026) | $\|\mathbf{S} - \mathbf{D}_t\mathbf{S}_{t-1}\|_F^2 - 2\langle\mathbf{S}^\top\mathbf{k}_t, \mathbf{w}_t\mathbf{v}_t - (\mathbf{D}_t\mathbf{S}_{t-1})^\top\mathbf{b}_t\mathbf{k}_t\rangle$ | $\mathbf{S}_t = (\mathbf{I} - \mathbf{k}_t(\mathbf{b}_t\mathbf{k}_t)^\top)\mathbf{D}_t\mathbf{S}_{t-1} + \mathbf{k}_t(\mathbf{w}_t\mathbf{v}_t)^\top$ | $\mathbf{D}_t, \mathbf{b}_t, \mathbf{w}_t$ |
| *Bayesian filtering write* | | | |
| KLA (Ours) | $\underbrace{\mathbf{\Lambda}_t^{\text{prior}}\|\mathbf{S} - \bar{\mathbf{A}}\mathbf{S}_{t-1}\|_F^2 + \mathbf{\Lambda}_t^v\|\mathbf{S}^\top\mathbf{k}_t - \mathbf{v}_t\|^2}_{\text{uncertainty- (precision-) weighted least squares}}$ | $\mathbf{S}_t = \bar{\mathbf{A}}\Big(\mathbf{I} - \dfrac{\mathbf{k}_t^2(\mathbf{\Lambda}_t^v)^\top}{\mathbf{\Lambda}_t}\Big)\mathbf{S}_{t-1} + \dfrac{\mathbf{k}_t(\mathbf{\Lambda}_t^v\mathbf{v}_t)^\top}{\mathbf{\Lambda}_t}$ | $\bar{\mathbf{A}}, \bar{\mathbf{P}}^\ddagger, \mathbf{\Lambda}_t^v$, Möbius recursion $(\mathbf{\Lambda}_t)$ |

*Table 6.* Local online objectives and state updates across linear-attention mixers. Orange marks the forget gate (coefficient of $\mathbf{S}_{t-1}$), green the write gate (coefficient of $\mathbf{k}_t\mathbf{v}_t^\top$), and blue the quantities unique to KLA, the precision weights in the objective (the only *precision-weighted* least-squares in the table, as marked by the underbrace) and the Möbius recursion supplying $\mathbf{\Lambda}_t$. Subscript $t$ denotes per-token learned quantities; symbols without subscript denote global parameters. KLA differs structurally by deriving its gating from a coupled recursion, the matrix-Möbius precision update of Theorem 1, whose solution $\mathbf{\Lambda}_t$ supplies the precision-ratio gain visible in both forget and write terms.

## E. KLA as Online Learning: A Precision-Weighted Regression View

Recent sub-quadratic mixers are increasingly unified through an *online-learning* / fast-weight lens, first articulated for SSMs by Longhorn (Liu et al., 2025) and developed in follow-up linear-attention and delta-rule work (Yang et al., 2023; 2024; Katharopoulos et al., 2020; Dao & Gu, 2024): each recurrent update is the closed-form minimiser of a per-token objective that trades a *proximal* term (stay close to the decayed memory) against a *fit* term (absorb the current token evidence).

Reading the information-mean recursion of Theorem 2 in moment coordinates, $\boldsymbol{\mu}_t = \boldsymbol{\eta}_t \oslash \boldsymbol{\lambda}_t$, collapses the predict/update pair into a single gated linear recurrence-the diagonal/scalar instance of the KLA row of Table 6:

**Corollary 3** (Posterior-mean recursion as a gated RNN update). *The moment-form posterior mean* $\boldsymbol{\mu}_t = \boldsymbol{\eta}_t \oslash \boldsymbol{\lambda}_t$ *of the information recursion in Theorem 2 satisfies the gated recurrence*

$$\boldsymbol{\mu}_t = \bar{\mathbf{a}}_t \odot \Big(1 - \frac{\mathbf{k}_t^2 \odot \mathbf{\Lambda}_t^{\text{v}}}{\boldsymbol{\lambda}_t}\Big) \odot \boldsymbol{\mu}_{t-1} + \frac{\mathbf{k}_t \odot \mathbf{\Lambda}_t^{\text{v}} \odot \mathbf{v}_t}{\boldsymbol{\lambda}_t}, \tag{23}$$

*the diagonal/scalar instance of the* KLA *state update in Table 6.*

We prove Corollary 3 in Section F, using $\boldsymbol{\lambda}_t^{\text{prior}} = \boldsymbol{\rho}_t \odot \boldsymbol{\lambda}_{t-1}$ from Step 1 of Theorem 1. This $\boldsymbol{\mu}_t$ is the exact minimiser $\arg\min_{\boldsymbol{\mu}} L_t$ of the precision-weighted least-squares objective

$$L_t(\boldsymbol{\mu}) = \mathbf{\Lambda}_t^{\text{v}}\left\|\mathbf{v}_t - \mathbf{k}_t \odot \boldsymbol{\mu}\right\|^2 + \boldsymbol{\rho}_t \odot \boldsymbol{\lambda}_{t-1}\left\|\boldsymbol{\mu} - \bar{\mathbf{a}}_t \odot \boldsymbol{\mu}_{t-1}\right\|^2, \tag{24}$$

balancing a fit term (weighted by the observation precision $\mathbf{\Lambda}_t^{\text{v}}$) against a proximal term anchoring $\boldsymbol{\mu}$ to the propagated mean $\bar{\mathbf{a}}_t \odot \boldsymbol{\mu}_{t-1}$ with weight the prior precision $\boldsymbol{\lambda}_t^{\text{prior}} = \boldsymbol{\rho}_t \odot \boldsymbol{\lambda}_{t-1}$, the history-dependent factor $\boldsymbol{\rho}_t$ of Equation (17), i.e. the prior-to-previous precision ratio. Table 6 places KLA in this template alongside representative delta-rule and gated mixers; setting $\bar{\mathbf{p}}_t = 0$ at constant precision freezes this gate and recovers a Mamba-2-style fixed-decay recurrence.

---

*For two matrices $\mathbf{A}, \mathbf{B}$ of equal shape we overload the fraction $\mathbf{A}/\mathbf{B}$ to denote *element-wise* (Hadamard) division; products between vectors of equal dimension (e.g. $\mathbf{b}_t\mathbf{k}_t, \mathbf{w}_t\mathbf{v}_t, \mathbf{\Lambda}_t^v\mathbf{v}_t$) are element-wise. Vector outer products such as $\mathbf{k}_t\mathbf{v}_t^\top$ and $\mathbf{k}_t^2(\mathbf{\Lambda}_t^v)^\top$ form $d_k \times d_v$ matrices in the standard way.

‡The process-noise parameter $\bar{\mathbf{P}}$ does not appear explicitly in the state update; it enters the model only *implicitly* through the matrix-Möbius precision recursion of Thm. 1, via the state-dependent denominator $\rho_t = 1/(\bar{\mathbf{A}}^2 + \bar{\mathbf{P}}\mathbf{\Lambda}_{t-1})$. Setting $\bar{\mathbf{P}} = 0$ collapses $\rho_t$ to a constant and reduces KLA to a recursive least-squares variant (von Oswald et al., 2025; Peng et al., 2026) with a linear update in place of the Möbius-structured nonlinear update.

# F. Theorems and Proofs

**Theorem 1** (Precision Update as a Möbius Transformation). *Let $\boldsymbol{\lambda}_t$ be the posterior precision at time $t$ in the diagonal linear-Gaussian model Equation* (7)-*Equation* (9). *Define $\boldsymbol{\phi}_t := \mathbf{k}_t^2 \odot \boldsymbol{\Lambda}_t^{\mathrm{v}}$. Then the map $\boldsymbol{\lambda}_{t-1} \mapsto \boldsymbol{\lambda}_t$ is a linear-fractional (Möbius) transform:*

$$\boldsymbol{\lambda}_t = \mathbf{M}_t(\boldsymbol{\lambda}_{t-1}) = \frac{\boldsymbol{\alpha}_t \odot \boldsymbol{\lambda}_{t-1} + \boldsymbol{\beta}_t}{\boldsymbol{\gamma}_t \odot \boldsymbol{\lambda}_{t-1} + \boldsymbol{\delta}_t}, \tag{15}$$

$$\mathbf{M}_t = \begin{pmatrix} \boldsymbol{\alpha}_t & \boldsymbol{\beta}_t \\ \boldsymbol{\gamma}_t & \boldsymbol{\delta}_t \end{pmatrix} = \begin{pmatrix} 1 + \bar{\mathbf{p}}_t \odot \boldsymbol{\phi}_t & \bar{\mathbf{a}}_t^2 \odot \boldsymbol{\phi}_t \\ \bar{\mathbf{p}}_t & \bar{\mathbf{a}}_t^2 \end{pmatrix}. \tag{16}$$

---

*Proof of Theorem 1.* Consider the 1D (or diagonal) linear-Gaussian state space model

$$z_t = a_t z_{t-1} + \varepsilon_t, \qquad \varepsilon_t \sim \mathcal{N}(0, p_t), \qquad v_t = k_t z_t + \nu_t, \qquad \nu_t \sim \mathcal{N}(0, (\Lambda_t^{\mathrm{v}})^{-1}),$$

and define the posterior precision $\lambda_t := \mathrm{Var}(z_t \mid v_{1:t})^{-1}$.

**Step 1 (Information-form predict/update).**

$$(\text{predict}) \quad \lambda_t^{\mathrm{prior}} = \left(a_t^2 \lambda_{t-1}^{-1} + p_t\right)^{-1} = \frac{\lambda_{t-1}}{a_t^2 + p_t \lambda_{t-1}}, \qquad (\text{update}) \quad \lambda_t = \lambda_t^{\mathrm{prior}} + k_t^2 \Lambda_t^{\mathrm{v}}.$$

**Step 2 (Single-step recursion and rearrangement).** Eliminate the intermediate $\lambda_t^{\mathrm{prior}}$ to obtain

$$\lambda_t = \frac{\lambda_{t-1}}{a_t^2 + p_t \lambda_{t-1}} + k_t^2 \Lambda_t^{\mathrm{v}} = \frac{(1 + p_t k_t^2 \Lambda_t^{\mathrm{v}}) \lambda_{t-1} + a_t^2 k_t^2 \Lambda_t^{\mathrm{v}}}{p_t \lambda_{t-1} + a_t^2}.$$

Hence $\lambda_t$ is a linear-fractional (Möbius) transform of $\lambda_{t-1}$,

$$\lambda_t = M_t(\lambda_{t-1}) = \frac{\alpha_t \lambda_{t-1} + \beta_t}{\gamma_t \lambda_{t-1} + \delta_t}, \qquad M_t = \begin{pmatrix} \alpha_t & \beta_t \\ \gamma_t & \delta_t \end{pmatrix} = \begin{pmatrix} 1 + p_t \phi_t & a_t^2 \phi_t \\ p_t & a_t^2 \end{pmatrix},$$

where $\phi_t := k_t^2 \Lambda_t^{\mathrm{v}}$. Thus, the precision recursion is a Möbius transformation with the stated matrix form. The diagonal multivariate case follows by applying the same scalar derivation to each diagonal element. $\square$

---

**Corollary 1** (Precision Updates via Parallel Prefix Scan). *Given $\boldsymbol{\lambda}_0$ and matrices $\{\mathbf{M}_t\}_{t=1}^T$ from Theorem 1, the posterior precisions $\{\boldsymbol{\lambda}_t\}_{t=1}^T$ can be computed by a parallel prefix scan over $\{\mathbf{M}_t\}$ with $\mathcal{O}(T)$ work and $\mathcal{O}(\log T)$ depth.*

---

*Proof of Corollary 1.* By Theorem 1, each precision update is a Möbius transformation: $\lambda_t = M_t(\lambda_{t-1})$. Since Möbius transformations compose via matrix multiplication, the precision at time $t$ can be expressed as

$$\lambda_t = M_t \circ M_{t-1} \circ \cdots \circ M_1(\lambda_0) = (M_t \cdot M_{t-1} \cdots M_1) \cdot \lambda_0,$$

where the composition $M_{1:t} := \prod_{s=1}^t M_s$ is computed via standard $2 \times 2$ matrix multiplication. Since matrix multiplication is associative, the product $M_{1:T}$ can be computed via a parallel prefix scan (Blelloch, 1990) with $\mathcal{O}(T)$ work and $\mathcal{O}(\log T)$ depth on $T$ processors. Once the prefix products $\{M_{1:t}\}_{t=1}^T$ are available, each precision $\lambda_t = M_{1:t} \cdot \lambda_0$ is obtained in $\mathcal{O}(1)$ time, yielding the claimed complexity. $\square$

---

**Remark** (Practical Implementation). *In practice, the parallel prefix scan can be implemented efficiently on modern hardware using frameworks such as JAX's* `lax.associative_scan` *or PyTorch's parallel primitives. The $\mathcal{O}(\log T)$ depth translates to logarithmic wall-clock time on sufficiently parallel hardware, matching the computational efficiency of deterministic SSMs like Mamba while maintaining probabilistic semantics.*

**Theorem 2** (Mean Update as Affine Transformations). *Let $\boldsymbol{\eta}_t := \boldsymbol{\lambda}_t \odot \boldsymbol{\mu}_t$ be the posterior information mean and $\boldsymbol{\Lambda}_t^{\mathrm{v}}$ the value precision. Given the precision path $\{\boldsymbol{\lambda}_t\}$, the information mean evolves affinely:*

$$\boldsymbol{\eta}_t = \underbrace{(\bar{\mathbf{a}}_t^2 + \bar{\mathbf{p}}_t \odot \boldsymbol{\lambda}_{t-1})^{-1} \odot \bar{\mathbf{a}}_t}_{\mathbf{f}_t \text{ (history-dependent forget gate)}} \odot \boldsymbol{\eta}_{t-1} \; + \; \underbrace{\mathbf{k}_t \odot \boldsymbol{\Lambda}_t^{\mathrm{v}} \odot \mathbf{v}_t}_{\text{token evidence}}. \tag{18}$$

*Proof of Theorem 2.* Write the information parameters as $\eta_t := \Lambda_t \mu_t$ and let $\Lambda_t^{\mathrm{v}}$ denote the value precision. The information-form measurement update is

$$\Lambda_t \; = \; \Lambda_t^{\mathrm{prior}} + k^2 \Lambda_t^{\mathrm{v}}, \qquad \eta_t \; = \; \eta_t^{\mathrm{prior}} + k \, \Lambda_t^{\mathrm{v}} \, v_t,$$

so the observation contribution is affine with coefficient $k \, \Lambda_t^{\mathrm{v}}$.

For the time-prediction of the information mean, use $\mu_t^{\mathrm{prior}} = a \, \mu_{t-1}$ and $\eta_t^{\mathrm{prior}} = \Lambda_t^{\mathrm{prior}} \mu_t^{\mathrm{prior}}$. Eliminating the means gives

$$\eta_t^{\mathrm{prior}} \; = \; \Lambda_t^{\mathrm{prior}} \, a \, \mu_{t-1} \; = \; \Lambda_t^{\mathrm{prior}} \, a \, \Lambda_{t-1}^{-1} \, \eta_{t-1}.$$

Hence, given the (known) precision path $\{\Lambda_t, \Lambda_t^{\mathrm{prior}}\}$,

$$\eta_t \; = \; \underbrace{\left(\Lambda_t^{\mathrm{prior}} \, a \, \Lambda_{t-1}^{-1}\right)}_{f_t} \eta_{t-1} \; + \; k \, \Lambda_t^{\mathrm{v}} \, v_t, \qquad \mu_t \; = \; \Lambda_t^{-1} \eta_t.$$

Finally, substituting the information-form predict step $\Lambda_t^{\mathrm{prior}} = \Lambda_{t-1}/(a^2 + p \, \Lambda_{t-1})$ (Step 1 of the proof of Theorem 1) eliminates $\Lambda_t^{\mathrm{prior}}$ from the forget gate,

$$f_t \; = \; \Lambda_t^{\mathrm{prior}} \, a \, \Lambda_{t-1}^{-1} \; = \; \frac{a}{a^2 + p \, \Lambda_{t-1}} \; = \; \left(a^2 + p \, \Lambda_{t-1}\right)^{-1} a,$$

which is exactly the forget gate in the statement of Theorem 2. This proves the claimed affine form. $\square$

**Corollary 3** (Posterior-mean recursion as a gated RNN update). *The moment-form posterior mean $\boldsymbol{\mu}_t = \boldsymbol{\eta}_t \oslash \boldsymbol{\lambda}_t$ of the information recursion in Theorem 2 satisfies the gated recurrence*

$$\boldsymbol{\mu}_t = \bar{\mathbf{a}}_t \odot \left(1 - \frac{\mathbf{k}_t^2 \odot \boldsymbol{\Lambda}_t^{\mathrm{v}}}{\boldsymbol{\lambda}_t}\right) \odot \boldsymbol{\mu}_{t-1} \; + \; \frac{\mathbf{k}_t \odot \boldsymbol{\Lambda}_t^{\mathrm{v}} \odot \mathbf{v}_t}{\boldsymbol{\lambda}_t}, \tag{23}$$

*the diagonal/scalar instance of the* KLA *state update in Table 6.*

*Proof of Corollary 3.* By Theorem 2, $\eta_t = f_t \, \eta_{t-1} + k_t \, \Lambda_t^{\mathrm{v}} \, v_t$ with forget gate $f_t = \lambda_t^{\mathrm{prior}} \, a_t \, \lambda_{t-1}^{-1}$. By Step 1 of the proof of Theorem 1, the predicted precision is

$$\lambda_t^{\mathrm{prior}} \; = \; \frac{\lambda_{t-1}}{a_t^2 + p_t \, \lambda_{t-1}} \; = \; \rho_t \, \lambda_{t-1}, \qquad \lambda_t \; = \; \lambda_t^{\mathrm{prior}} + k_t^2 \, \Lambda_t^{\mathrm{v}}.$$

Substituting $\eta_{t-1} = \lambda_{t-1} \, \mu_{t-1}$ and $\mu_t = \eta_t/\lambda_t$, and using $f_t \, \lambda_{t-1} = \lambda_t^{\mathrm{prior}} \, a_t$,

$$\mu_t \; = \; \frac{\lambda_t^{\mathrm{prior}} \, a_t \, \mu_{t-1} + k_t \, \Lambda_t^{\mathrm{v}} \, v_t}{\lambda_t}.$$

Since $\lambda_t^{\mathrm{prior}} = \lambda_t - k_t^2 \, \Lambda_t^{\mathrm{v}}$, we have $\lambda_t^{\mathrm{prior}}/\lambda_t = 1 - k_t^2 \, \Lambda_t^{\mathrm{v}}/\lambda_t$, hence

$$\mu_t \; = \; a_t \left(1 - \frac{k_t^2 \, \Lambda_t^{\mathrm{v}}}{\lambda_t}\right) \mu_{t-1} + \frac{k_t \, \Lambda_t^{\mathrm{v}}}{\lambda_t} \, v_t,$$

the claimed gated recurrence. The diagonal multivariate case applies elementwise. $\square$

**Theorem 3** (Convolutional Form for Deterministic LTI Systems). *Under the conditions: (i) time-invariant dynamics, $\mathbf{a}_t \equiv \mathbf{a}$, $\mathbf{k}_t \equiv \mathbf{k}$; and (ii) deterministic dynamics, process noise $\mathbf{p}_t = \mathbf{0}$, both the precision and information mean updates reduce to block-Toeplitz convolutions computable in $\mathcal{O}(T \log T)$ time via FFT/NTT:*

**Precision updates:** $\quad \mathbf{\Lambda}_t = \sum_{s=0}^{t} \mathbf{a}^{-2(t-s)} \odot \mathbf{k}^2 \odot \mathbf{\Lambda}_s^{\mathrm{v}}$

**Information mean updates:** $\quad \boldsymbol{\eta}_t = \mathbf{k} \odot \sum_{s=0}^{t} \mathbf{a}^{t-s} \odot \mathbf{\Lambda}_s^{\mathrm{v}} \odot \mathbf{v}_s, \qquad \boldsymbol{\mu}_t = \mathbf{\Lambda}_t^{-1} \odot \boldsymbol{\eta}_t$

---

*Proof of Theorem 3.* When $p_t = 0$ (deterministic dynamics), the Möbius transformation from Theorem 1 simplifies. Specifically, with zero process noise, the precision recursion becomes $\lambda_t = a^{-2}\lambda_{t-1} + k^2\Lambda_t^{\mathrm{v}}$. For time-invariant $a$ and $k$, this unrolls to:

$$\lambda_t = a^{-2t}\lambda_0 + k^2 \sum_{s=0}^{t} a^{-2(t-s)}\Lambda_s^{\mathrm{v}}$$

which is a discrete convolution with kernel $\kappa[n] = k^2 a^{-2n}$.

Similarly, from Theorem 2, when $p_t = 0$ we have $f_t = a^{-1}$ constant, so the information mean recursion $\eta_t = a^{-1}\eta_{t-1} + k\Lambda_t^{\mathrm{v}}v_t$ unrolls to:

$$\eta_t = k \sum_{s=0}^{t} a^{t-s} \Lambda_s^{\mathrm{v}} v_s$$

which is also a convolution with kernel $h[n] = k\, a^n$.

Both convolutions have Toeplitz structure and can be computed via FFT in $\mathcal{O}(\log T)$ parallel time using the convolution theorem. $\qquad\square$

---

**Remark** (Practical Note). *While theoretically interesting, the convolutional form is primarily relevant for perfectly deterministic systems ($\mathbf{p} = \mathbf{0}$). In practice,* KLA *uses the more general parallel scan formulation (Corollary 1) which handles stochastic dynamics ($\mathbf{p}_t > \mathbf{0}$) efficiently.*

**Corollary 2** (Mean Updates via Parallel Prefix Scan). *The posterior information means $\{\boldsymbol{\eta}_t\}_{t=1}^T$ are computable via parallel prefix scan over the affine transformations in Theorem 2 in $\mathcal{O}(T)$ work and $\mathcal{O}(\log T)$ depth.*

*Proof of Corollary 2.* From Theorem 2, the information mean update has the affine form:

$$\eta_t = f_t \eta_{t-1} + k_t \Lambda_t^{\mathrm{Y}} v_t$$

where $f_t = \frac{a_t}{a_t^2 + p_t \lambda_{t-1}}$ (which simplifies to $a^{-1}$ when $p_t = 0$).

Define the binary associative operator $\oplus$ on pairs $(f, b)$ by:

$$(f_2, b_2) \oplus (f_1, b_1) = (f_2 f_1, f_2 b_1 + b_2)$$

This is the standard associative operator for affine transformations.

The composed transformation at time $t$ is:

$$(f_{1:t}, b_{1:t}) = (f_t, k_t \Lambda_t^{\mathrm{Y}} v_t) \oplus (f_{t-1}, k_{t-1} \Lambda_{t-1}^{\mathrm{Y}} v_{t-1}) \oplus \cdots \oplus (f_1, k_1 \Lambda_1^{\mathrm{Y}} v_1)$$

which gives $\eta_t = f_{1:t} \eta_0 + b_{1:t}$.

Since $\oplus$ is associative, the sequence of compositions can be computed via a parallel prefix scan (Blelloch, 1990) with $\mathcal{O}(T)$ work and $\mathcal{O}(\log T)$ depth. Once all information means $\{\eta_t\}_{t=1}^T$ are computed, the posterior means are obtained by normalising with the precision path from Corollary 1: $\mu_t = \Lambda_t^{-1} \eta_t$. $\qquad \square$

# G. Additional Empirical Results

## G.1. Ablation on the Importance of Process Noise Parameter

We conduct an ablation study motivated by the theoretical efficiency of fixing the process noise parameter $\mathbf{p}_t = 0$ and the observation operator $k_t$. Under these constraints, the system reduces to a linear time-invariant (LTI) model that can be computed via convolution and unrolled in $\mathcal{O}(\log T)$ time using FFT (see Theorem 3). This ablation investigates whether such computational advantages translate to practical benefits.

Figure 7 compares KLA with learnable process noise against a variant where process noise is fixed to zero (deterministic dynamics, but retaining observation variance $(\Lambda^v)^{-1}$). The results show that removing process noise leads to significantly worse performance across all MADLAB tasks, with accuracy degrading by an average of 49.8 percentage points. The degradation is particularly severe on memorisation (93.6% drop) and in-context recall tasks (54.7% drop), while selective copying shows the smallest decline (18.7% drop).

These empirical findings suggest that process noise is crucial for maintaining model performance, despite the computational overhead of the full Kalman filter formulation. A theoretical investigation of the stability properties of the system, particularly regarding Riccati controllability and observability, and the role of $\mathbf{p}_t$ in system stabilisation, represents an interesting direction for future work.

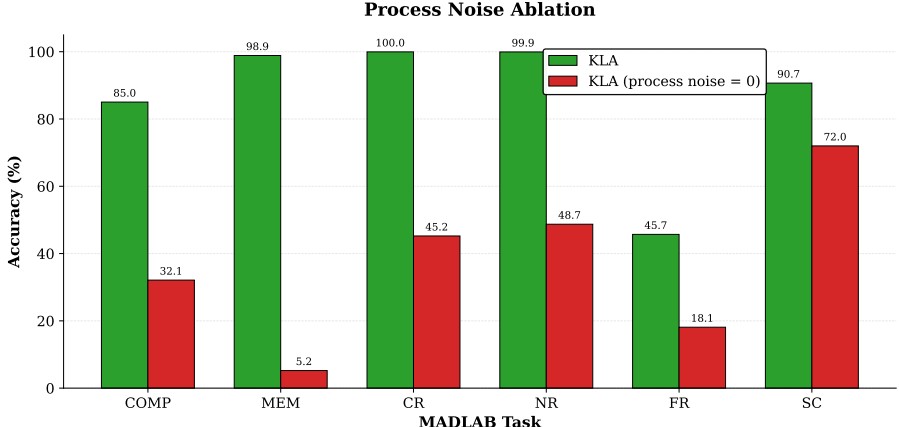

*Figure 7.* **Process noise ablation on MADLAB tasks.** Comparison of KLA with learnable process noise versus a variant with process noise fixed to zero ($\mathbf{p}_t = 0$). Removing process noise leads to severe performance degradation, with an average accuracy drop of 49.8 percentage points.

## G.2. Long-Context Associative Recall (Full Sweep)

Multi-Query Associative Recall (MQAR) (Arora et al., 2023) is particularly demanding for fixed-size recurrent architectures, as it directly tests their capacity to store and retrieve multiple key-value associations, a fundamental bottleneck often limited by state dimensionality. Following the Zoology benchmark (Arora et al., 2023), we evaluate on a considerably harder configuration than typically studied: sequence length $T = 2048$ with vocabulary size $V = 256$ (matching or exceeding model dimensions; see Table 9).

Figure 8 reveals distinct scaling behaviours across architectures. KLA consistently outperforms Mamba across all dimensions and substantially outperforms GLA, which fails to learn the task under this extreme setting. At lower dimensions ($d = 64, 128$), GDN performs best; its delta-rule mechanism is specifically designed for strong associative recall at limited capacity (Yang et al., 2024), and it operates with a slightly higher effective state size (see Table 12). At $d = 256$, KLA outperforms all baselines and reaches near-perfect accuracy ($> 95\%$).

We hypothesise that uncertainty-weighted updates help mitigate state "saturation" under heavy key-value load. Unlike purely additive accumulators, KLA's fractional-linear (Möbius) precision updates (Equation (17)) adaptively down-weight observations based on posterior uncertainty: when the latent state becomes saturated with many associations, high-confidence historical information naturally suppresses unreliable new observations. This provides an implicit, learned compression mechanism, analogous to selective pattern storage in high-capacity associative memories (Storkey & Valabregue, 1997), without requiring explicit dimension scaling.

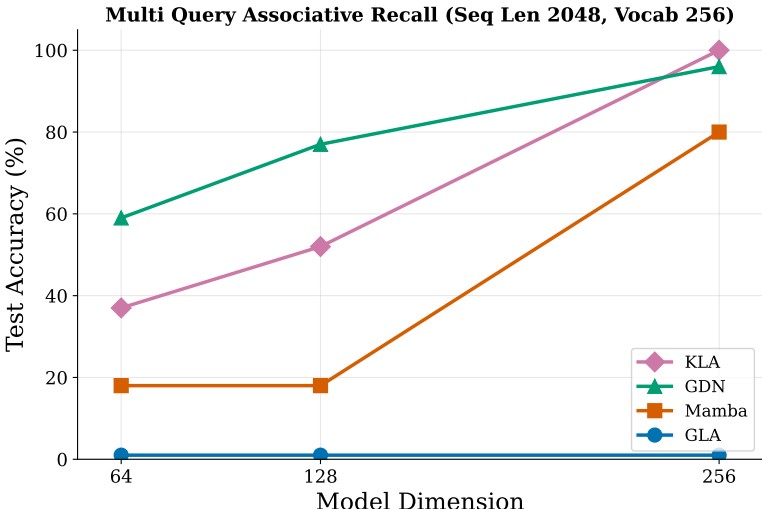

*Figure 8.* MQAR accuracy vs. model dimension ($T = 2048$, $V = 256$). KLA achieves near-perfect accuracy at dimension 256, outperforming GDN and Mamba. GLA fails to learn the task in this challenging setting.

### G.3. Runtime Scaling

**Setup.** We benchmark forward-only runtimes for three KLA implementations: a recurrent (time-stepped) Kalman update, the built-in Torch associative scan, and a custom Triton associative scan kernel. All measurements use Torch 2.9.1 on an NVIDIA A100 (80GB), hidden size 960, and float32 precision. The forward-only pass uses a batch size of 1, simulating test-time prompt processing for a 182 million parameter model-critical for deploying sequence models in production settings.

**Results.** Figure 9 shows that the scan-based implementations scale efficiently with sequence length compared to the recurrent counterparts. These results confirm that information-form filtering can be implemented with the same scan-parallel profile as modern SSM/GLA-style mixers, making KLA practical for long-context prompt processing.

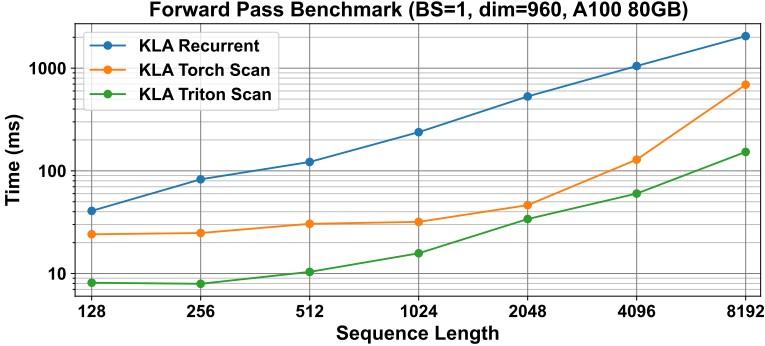

*Figure 9.* **Forward-only (prompt processing) runtime scaling.** Wall-clock runtime of KLA implementations across sequence lengths during the forward-only pass. Torch Scan uses `torch._higher_order_ops.associative_scan`; Triton Scan uses custom forward kernels.

### G.4. Equivalent Attention Matrix

As shown in Theorem 2, the information-mean recurrence $\eta_t = f_t\,\eta_{t-1} + k_t\,\Lambda_t^{\text{y}}\,v_t$ (Equation (18)) can be unrolled into a lower-triangular attention matrix $\mathbf{W}$ whose entries are products of history-dependent forget gates $f_s$, keys $k_j$, and observation precisions $\Lambda_j^{\text{y}}$. Folding in the output readout query $\mathbf{q}$ and posterior precision scaling $\boldsymbol{\lambda}^{-1}$ gives the full per-channel sequence transformation $\mathbf{M}_{\text{seq}} = \operatorname{diag}(\mathbf{q} \odot \boldsymbol{\lambda}^{-1})\,\mathbf{W}$, so that $\mathbf{y} = \mathbf{M}_{\text{seq}}\,\mathbf{v}$ (plus init-state terms). Figure 10 visualises this two-step structure.

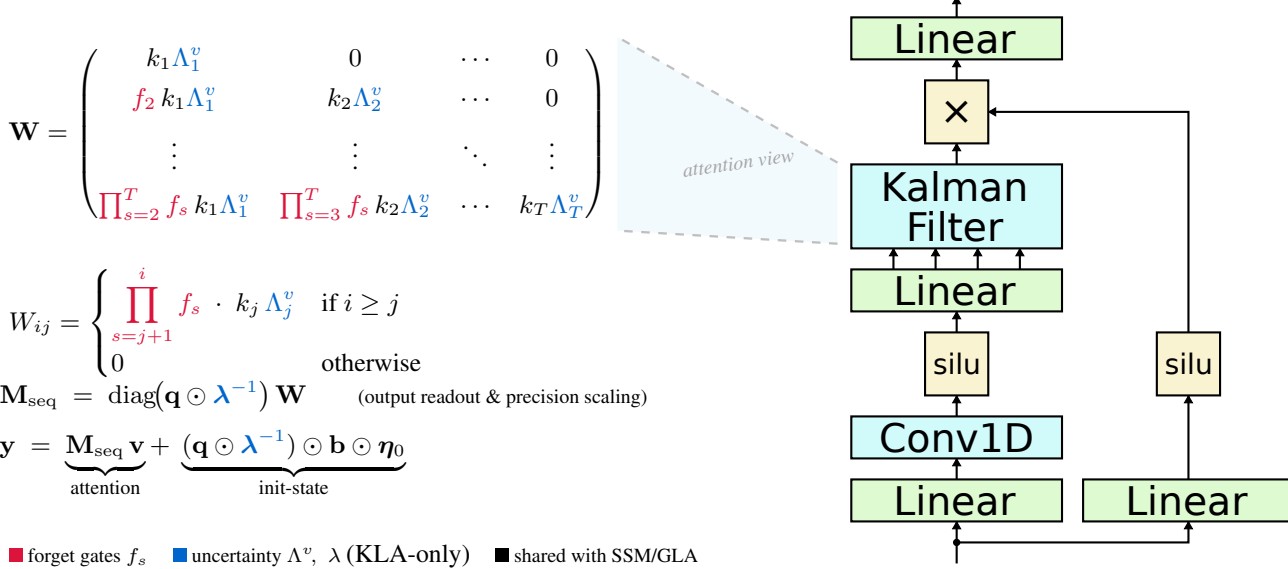

$$\mathbf{W} = \begin{pmatrix} k_1 \Lambda_1^v & 0 & \cdots & 0 \\ f_2\, k_1 \Lambda_1^v & k_2 \Lambda_2^v & \cdots & 0 \\ \vdots & \vdots & \ddots & \vdots \\ \prod_{s=2}^{T} f_s\, k_1 \Lambda_1^v & \prod_{s=3}^{T} f_s\, k_2 \Lambda_2^v & \cdots & k_T \Lambda_T^v \end{pmatrix}$$

$$W_{ij} = \begin{cases} \prod_{s=j+1}^{i} f_s \cdot k_j\, \Lambda_j^v & \text{if } i \geq j \\ 0 & \text{otherwise} \end{cases}$$

$$\mathbf{M}_{\text{seq}} = \text{diag}(\mathbf{q} \odot \boldsymbol{\lambda}^{-1})\, \mathbf{W} \qquad \text{(output readout \& precision scaling)}$$

$$\mathbf{y} = \underbrace{\mathbf{M}_{\text{seq}}\, \mathbf{v}}_{\text{attention}} + \underbrace{(\mathbf{q} \odot \boldsymbol{\lambda}^{-1}) \odot \mathbf{b} \odot \boldsymbol{\eta}_0}_{\text{init-state}}$$

■ forget gates $f_s$    ■ uncertainty $\Lambda^v$, $\lambda$ (KLA-only)    ■ shared with SSM/GLA

*Figure 10.* **Block architecture and equivalent attention form.** *Left:* The block architecture follows the fused-MLP design of Mamba, with the Kalman Filter as a drop-in replacement for any SSM/Attention primitive. *Right:* Unrolling the information-mean recurrence (Equation (18): $\eta_t = f_t\, \eta_{t-1} + k_t\, \Lambda_t^v\, v_t$) yields a lower-triangular matrix $\mathbf{W}$ whose entries are products of history-dependent forget gates $f_s$, keys $k_j$, and observation precisions $\Lambda_j^v$. Applying the output readout query $q_i$ and posterior precision $\lambda_i^{-1}$ gives the full sequence transformation $\mathbf{M}_{\text{seq}} = \text{diag}(\mathbf{q} \odot \boldsymbol{\lambda}^{-1})\, \mathbf{W}$, the precision terms being unique to KLA and absent in standard SSMs/GLA.

### G.5. Kalman Attention Map Visualisation

Unrolling the information-mean recurrence equation 18 yields the lower-triangular matrix $\mathbf{W}$ described in Section G.4; folding in the readout query $\mathbf{q}_t$ and precision scaling $\boldsymbol{\lambda}^{-1}$ gives the per-channel sequence transformation $\mathbf{M}_{\text{seq}} = \text{diag}(\mathbf{q} \odot \boldsymbol{\lambda}^{-1})\, \mathbf{W}$ (cf. Figure 10). Because $\mathbf{M}_{\text{seq}}$ is lower-triangular, it has the same causal structure as a standard attention matrix; we therefore call it the *Kalman Attention Matrix*.

Figures 11 to 13 visualise the Kalman attention patterns learned by KLA on three MADLAB tasks, each showing four randomly chosen channels. The attention weight at position $(t, s)$ reflects how much the posterior mean at step $t$ is influenced by the observation at step $s$, weighted by the learned precision ratio.

These visualisations demonstrate that KLA learns task-appropriate attention patterns: selective retrieval for copying tasks, channel specialisation for recall, and uniform retention for memorization-all while maintaining the causal structure inherent to the Kalman filter formulation.

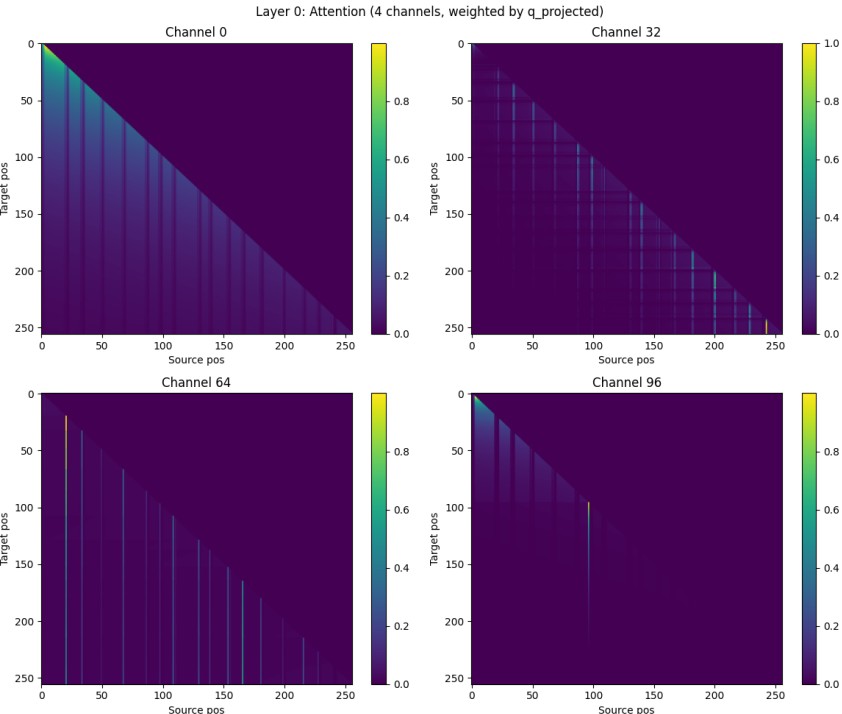

*Figure 11.* Attention maps for **Selective Copying** (sequence length 256). The model learns sparse, intermittent vertical bands, attending strongly to a small set of task-relevant positions (mostly copy positions). They either activate or suppress these relevant positions. Differences across channels indicate channel-level specialisation in which positions are retrieved.

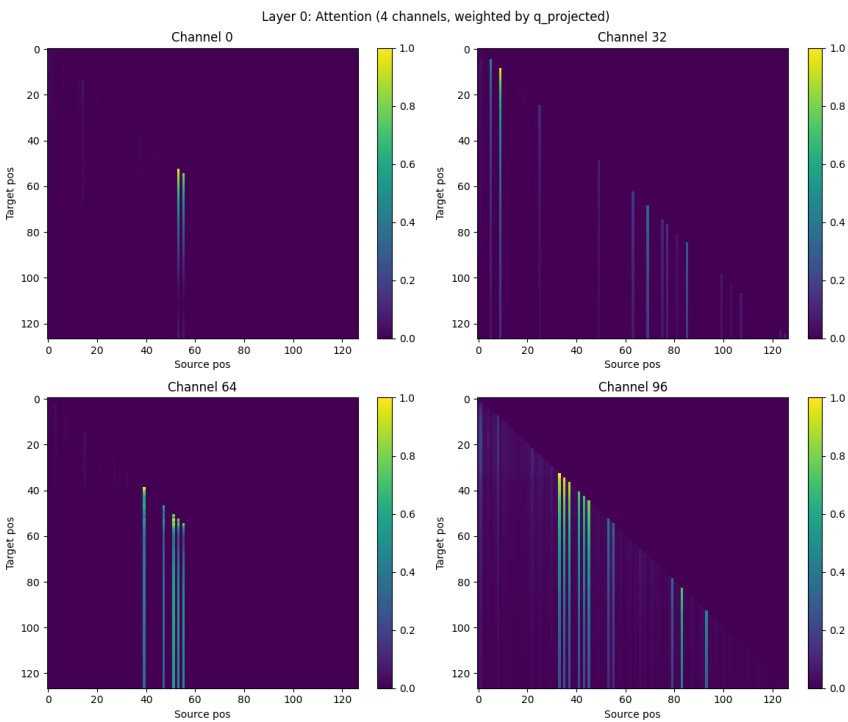

*Figure 12.* Attention maps for **In-Context Recall** (sequence length 128). The patterns are sparse and "pointer-like": within each channel, attention concentrates on a small set of source positions (vertical bands) across many target steps, with a mostly low-activation background. Channels show specialisation trends.

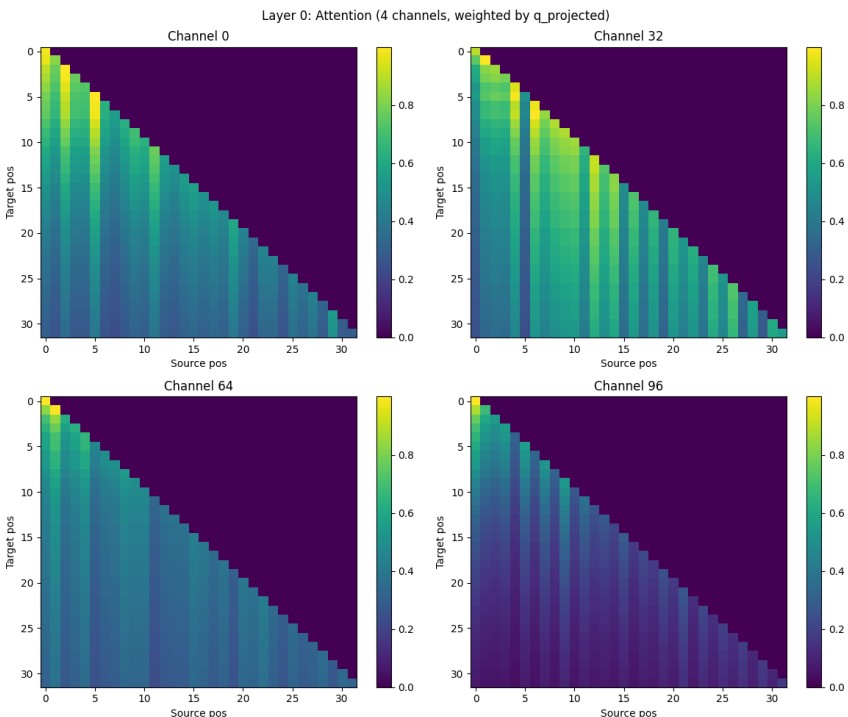

*Figure 13.* Attention maps for **Memorization** (sequence length 32). The smoother, more gradual decay reflects the task's requirement to maintain information uniformly across the sequence. All channels show similar patterns, indicating that memorization benefits from redundant, distributed storage rather than selective attention.

*Table 7.* Mapping from language skills/capabilities to the synthetic probe tasks. Green ✓ indicates the primary probes/tasks for these skills.

| Language Skill / Capability | In-context (MQAR) | Fuzzy Recall | Noisy Recall | Selective Copy | Com- pression | Memo- rization |
|---|---|---|---|---|---|---|
| **Prompt-level associative retrieval & in-context learning** *(bindings, multi-query, remapping, basic composition)* | ✓ | ✓ | ✓ | ✓ | ✗ | ✗ |
| **Selective attention & noise robustness** *(noise filtering, long-context stability, calibration)* | ✗ | ✗ | ✓ | ✓ | ✗ | ✗ |
| **Ordered copying & pointer-like control flow** *(order-sensitive working memory)* | ✗ | ✗ | ✗ | ✓ | ✗ | ✗ |
| **Information aggregation/bottlenecking** *(token concatenation for downstream decoding)* | ✗ | ✗ | ✗ | ✗ | ✓ | ✗ |
| **Parametric memory of stable facts** *(weight-based knowledge)* | ✗ | ✗ | ✗ | ✗ | ✗ | ✓ |

# H. Datasets and Benchmarks Used

## H.1. MAD LM Suite

To evaluate KLA as a language-modelling primitive, we use a standardised suite of six discrete token manipulation tasks. We adopt the MAD framework (Poli et al., 2024), which consolidates synthetic tasks from prior work into a common specification of data generation, splits, and evaluation. Each task probes a distinct capability (associative retrieval, span compositionality, noise robustness, ordered copying, single-token information aggregation, and parametric memory), allowing us to disentangle model strengths and weaknesses across language skills (see Table 7).

We use the baseline configuration from the MAD framework for all six tasks. Table 8 summarises the key parameters for each task.

*Table 8.* Baseline configurations for the six MAD tasks used in our experiments.

| Parameter | In-context (MQAR) | Fuzzy Recall | Noisy Recall | Selective Copy | Compression | Memorization |
|---|---|---|---|---|---|---|
| Vocab size | 16 | 16 | 16 | 16 | 16 | 256 |
| Training seqs | 12,800 | 12,800 | 12,800 | 12,800 | 12,800 | 256 |
| Sequence length | 128 | 128 | 128 | 256 | 32 | 32 |
| Noise tokens (%) | - | - | 20% | - | - | - |
| Key/value split | 8/8 | 8/8 | 8/8 | - | - | 128/128 |

**Additional task-specific settings:** Selective Copy (SC) uses `num_copy`=16; In-Context Recall (CR) and Fuzzy Recall (FR) use `multi_query`=True; FR uses motif sizes $k = v = 3$; Noisy Recall (NR) uses `noise_vocab`=16 with noise fraction 0.2.

### H.1.1. TASK DESCRIPTIONS

**In-context recall (MQAR).** Sequences consist of key-value pairs with separate vocabularies. The model must predict values for keys that appeared earlier in the sequence. Key-value mappings are randomly shuffled between sequences, forcing the model to learn in-context rather than memorising fixed associations.

**Fuzzy in-context recall.** An extension of in-context recall where keys and values are represented by variable-length spans (1-3 tokens). This tests the model's ability to handle compositional keys and maintain associations across multi-token representations.

**Noisy in-context recall.** Similar to in-context recall, but with 20% of tokens from a separate noise vocabulary randomly inserted. This evaluates the model's robustness to irrelevant information and selective attention capabilities.

**Selective copying.** Sequences contain random tokens interspersed with special [blank] and [insert] tokens. The model must copy non-special tokens to [insert] positions in order, while learning to ignore tokens near [blank] markers. This tests selective memorisation and order-preserving working memory.

**Compression.** Sequences of random tokens ending with a compression token [c]. The model must compress all sequence information into the representation at the compression token position, such that a fixed two-layer MLP can reconstruct any

input token given the compressed representation plus a positional encoding.

**Memorization.** A fixed key-value dictionary is used across all sequences. In each sequence, keys appear with their values masked by [insert] tokens. The model must learn the fixed mappings from the training data, as values never appear in the input during training.

### H.1.2. STANDARD TRAINING SETUP IN MAD FRAMEWORK

All tasks use the same architectural scaffolding and training procedure. Training hyperparameters are detailed in Section I.

For most tasks, the training objective is standard next-token prediction via log likelihood (cross-entropy loss). The **Compression** task uses a different setup: the model's output at the compression token position is passed through a separate two-layer MLP decoder (dimensions [240, 120]) along with a positional encoding to reconstruct each token in the sequence. During training, the model learns to compress the entire sequence into a single representation that enables accurate reconstruction.

All models use a single-layer architecture with a model dimension $d_{\text{model}} = 128$ and an effective state size of 2048.

## H.2. Long-Context MQAR

Table 9 lists the data configuration for the long-context MQAR benchmark task (Arora et al., 2023).

*Table 9.* **Long-Context MQAR data parameters.**

| Long-Context Multi-Query Associative Recall (MQAR) | | | | |
|---|---|---|---|---|
| **Setting** | **Seq. length** | **Vocab size** | **Training seqs** | **Eval seqs** |
| CR (hard) | 2,048 | 256 | 12,800 | 1,280 |

# I. Implementation Details

## I.1. Experimental Protocol

**Statistical significance.** We report results averaged over 5 random seeds to ensure statistical reliability. All comparisons between models use the same random seeds to ensure fairness. For all tasks, we report the mean performance across seeds. For the A5 state tracking task, we consider a task solved if the model achieves $\geq 90\%$ accuracy in at least one of the five seeds.

**Open-source code.** The complete implementation, including all custom Triton and CUDA kernels and experimental code, is publicly available at https://github.com/vaisakh-shaj/kalman-linear-attention to facilitate reproducibility and future research.

## I.2. Training Hyperparameters

We use default settings unless specifically stated otherwise. Table 10 lists the training configuration used across all experiments.

*Table 10.* **Training Hyperparameters** (fixed across all models and tasks)

| Hyperparameter | Value |
|---|---|
| Optimizer | AdamW |
| Learning rate | $1 \times 10^{-3}$ |
| Learning rate schedule | None |
| Maximum epochs | 750 |
| Early stopping patience | 70 epochs |
| Gradient clipping | 5.0 |
| Weight decay | 0.0 |
| Precision | 32-bit float |

**KLA-specific settings:** We use encoder MLP hidden dimension = 120, decoder MLP dimensions = [240, 120], process noise scale $\mathbf{p} = 0.01$ (initial value), discretisation timestep range $\Delta_t \in [0.001, 0.1]$, causal convolution kernel size = 4, and Monte Carlo samples = 10 (used for KLA probabilistic decoding; see Figure 4b).

## I.3. MAD-Lab Hyperparameters

To ensure fair comparison, all models are configured with equal effective state sizes by adjusting architecture-specific expansion parameters. For each model dimension, we match the total number of parameters used for state representation across architectures. Note that GatedDeltaNet does not allow exact state-size matching due to constraints on the minimum number of heads and the requirement that expanded key and value dimensions be multiples of 16 or 32. Hence, we assign GatedDeltaNet the next-largest feasible value compared to the other baselines. Table 11 shows the dimensional configuration for each architecture.

*Table 11.* **Model Dimensions** (configured for equal state size $S = 2048$)

| Model | # Blocks | $d_{\text{model}}$ | $d_{\text{state}}$ | Other | $S$ |
|---|---|---|---|---|---|
| KLA (Ours) | 1 | 128 | 8 | expand=1 | 2048 |
| Mamba | 1 | 128 | 16 | expand=1 | 2048 |
| GatedDeltaNet | 1 | 128 | - | $e_k$=0.5, $e_v$=1.0, $H$=4 | 2048 |
| GLA | 1 | 128 | - | $e_k$=0.5, $e_v$=1.0, $H$=4 | 2048 |
| mLSTM | 1 | 128 | - | $H$=16, proj_factor=2.0 | 2048 |

**Batch size:** We use a batch size of 172 for all MAD-Lab experiments.

## I.4. MQAR Hyperparameters

For the MQAR (hard) experiments, we evaluate across three model dimensions with equal state sizes per dimension. All models use 2 layers (repeated blocks). As in the MAD-Lab setup, GatedDeltaNet is given the nearest feasible configuration (see above). Table 12 lists the configuration for each architecture at each dimension.

*Table 12.* **MQAR Model Configurations** (state-matched per dimension)

| Model | # Blocks | $d_{\text{model}}$ | $d_{\text{state}}$ | Other | State Size |
|---|---|---|---|---|---|
| \multicolumn{6}{c}{$d = 64$ (batch size = 64)} | | | | | |
| KLA | 2 | 64 | 16 | expand=1 | 2,048 |
| Mamba | 2 | 64 | 16 | expand=2 | 2,048 |
| GatedDeltaNet | 2 | 64 | - | $e_k$=0.75, $e_v$=1.5, $H$=2 | 2,304 |
| GLA | 2 | 64 | - | $e_k$=1.0, $e_v$=1.0, $H$=4 | 2,048 |
| \multicolumn{6}{c}{$d = 128$ (batch size = 32)} | | | | | |
| KLA | 2 | 128 | 16 | expand=1 | 4,096 |
| Mamba | 2 | 128 | 16 | expand=2 | 4,096 |
| GatedDeltaNet | 2 | 128 | - | $e_k$=0.75, $e_v$=1.5, $H$=4 | 4,608 |
| GLA | 2 | 128 | - | $e_k$=0.5, $e_v$=1.0, $H$=4 | 4,096 |
| \multicolumn{6}{c}{$d = 256$ (batch size = 16)} | | | | | |
| KLA | 2 | 256 | 16 | expand=1 | 8,192 |
| Mamba | 2 | 256 | 16 | expand=2 | 8,192 |
| GatedDeltaNet | 2 | 256 | - | $e_k$=0.75, $e_v$=1.5, $H$=8 | 9,216 |
| GLA | 2 | 256 | - | $e_k$=0.25, $e_v$=1.0, $H$=4 | 8,192 |

## I.5. A5 State Tracking Hyperparameters

For the A5 state tracking experiments (Section 5.4), we use $d_{\text{model}} = 1024$ and $d_{\text{state}} = 16$ for KLA. We train for up to 500 epochs with early stopping (patience 50), and a learning rate of $3 \times 10^{-4}$, and we evaluate over 5 seeds. Values for the baselines are reported as in Merrill et al. (2024).

## I.6. Language-Modelling Pretraining and Evaluation Details

This appendix gives the pretraining and evaluation details for the language-modelling experiments of Section 5.5 (results in Table 3).

**Compute.** All language-modelling models were trained on NVIDIA H200 GPUs on the Isambard-AI National AI Research Resource (AIRR) (McIntosh-Smith et al., 2024).

**Pretraining corpus and objective.** All models are pretrained on FineWeb-Edu (Penedo et al., 2024) with the standard next-token-prediction (cross-entropy) objective in mixed precision (`bfloat16`, with the parallel scan layers kept in `fp32`). We train at two academic scales: a 45M-parameter model on 1.8B tokens and a 180M-parameter model on 10.9B tokens.

**Models and baselines.** Both the 45M and 180M models use a 12-layer backbone, and all architectures are parameter-matched at each scale. We compare four standalone mixers — GPT (softmax attention), Mamba, Gated DeltaNet (GDN), and KLA— and the hybrid GPT+KLA, in which *only the final attention layer* of the GPT backbone is replaced by a KLA block (Section 5.5). Baseline state/expansion settings are matched following the conventions in Section I.3; we report the KLA-specific per-scale configuration in Table 13.

**Optimisation.** Models are trained with AdamW and gradient clipping 3.0 (Section 5.6). The remaining optimisation settings — learning rate, schedule, warmup, batch size, and context length — are summarised in Table 13.

**Zero-shot evaluation.** We report zero-shot performance with the LM Evaluation Harness on eight commonsense benchmarks: LAMBADA (OpenAI split), HellaSwag, PIQA, ARC-Easy, ARC-Challenge, WinoGrande, OpenBookQA, and BoolQ (Table 3). We report accuracy (acc) for all tasks except HellaSwag and ARC-Challenge, for which we report length-normalised accuracy ($\text{acc}_n$).

*Table 13.* **KLA language-modelling pretraining configurations.** We report KLA-specific settings only. Both scales use a 12-layer KLA backbone. Baseline configurations are state-matched following Section I.3. All parameters are optimised with AdamW ($\beta = (0.8, 0.95)$, $\epsilon = 10^{-10}$); Muon is disabled. The per-group AdamW learning rate is scaled internally by $\sqrt{768/d_{\mathrm{model}}} \cdot \sqrt{B/2^{19}}$ (where $B$ is the token batch size); embeddings use a separate base rate of 0.3.

| KLA hyperparameter | 45M | 180M |
|---|---|---|
| Layers (depth) | 12 | 12 |
| $d_{\mathrm{model}}$ | 496 | 1360 |
| $d_{\mathrm{state}}$ | 32 | 32 |
| Causal conv. kernel size | 4 | 4 |
| Process-noise init $\mathbf{p}$ | 0.01 | 0.01 |
| Discretisation range $\Delta_t$ | [0.001, 0.1] | [0.001, 0.1] |
| Context length | 2048 | 2048 |
| Training tokens | 1.8B | 10.9B |
| Optimiser | AdamW | AdamW |
| Base learning rate (hidden/SSM) | $4 \times 10^{-3}$ | $4 \times 10^{-3}$ |
| Effective peak LR (hidden/SSM) | $2.5 \times 10^{-3}$ | $3.0 \times 10^{-3}$ |
| Embedding learning rate (base) | 0.3 | 0.3 |
| LR schedule | constant then linear decay (no warmup) | constant then linear decay (no warmup) |
| Warmup steps | 0 | 0 |
| Warmdown fraction | 0.4 | 0.4 |
| Weight decay (2D hidden/lm_head) | 0.1 | 0.1 |
| Weight decay (embed/1D) | 0.0 | 0.0 |
| Global batch size (tokens) | 131,072 | 524,288 |
| Gradient clipping | 3.0 | 3.0 |
| Precision | mixed bf16 (scan in fp32) | mixed bf16 (scan in fp32) |

