# OpenReview forum: "Kalman Linear Attention: Parallel Bayesian Filtering For Efficient Language Modeling and State Tracking"
_ICML.cc/2026/Conference — ICML 2026 regular_

### Official Review · Reviewer_jAQH · 2026-03-06

**Soundness:** 3
**Presentation:** 2
**Significance:** 3
**Originality:** 2
**Overall Recommendation:** 4
**Confidence:** 1

**Summary:**

The paper presents Kalman Linear Attention (KLA), a sequence modeling method that combines Kalman filtering and linear attention to create an efficient alternative to transformer attention. The authors show that these Kalman-style updates can be reformulated using a mathematical structure that allows parallel computation through prefix scans, enabling linear-time complexity similar to modern state-space models. Experiments on synthetic sequence tasks demonstrate that KLA performs well on different problems requiring memory, compression, and state tracking.

**Compliance With Llm Reviewing Policy:**

Affirmed.

**Key Questions For Authors:**

1 – How does Kalman Linear Attention differ conceptually and practically from state-space models (SSMs) and recurrent sequence models?

2 – Can the authors clarify the connection and differences between KLA and recent architectures that also use parallel recurrence?

3 – Can the authors provide runtime benchmarks and GPU memory comparisons with other linear attention methods?

**Limitations:**

yes

**Strengths And Weaknesses:**

Strengths: The main strength of the paper is its novel theoretical contribution, introducing a principled probabilistic framework for sequence modeling that explicitly tracks uncertainty in the hidden state. The reformulation that enables parallel Kalman filtering is mathematically elegant and makes the approach computationally efficient. The experimental results also show strong performance on tasks involving long-range dependencies and state tracking.

Weaknesses: The concept and novelty KLA has not been well-emphasized compared to state-space models (SSMs) and recurrent sequence models. The approach introduces additional complexity compared to simpler linear attention or state-space models, which may make training and scaling more challenging.

---

> ### Author Rebuttal · Authors · 2026-03-31
>
> We thank the reviewer for recognizing the theoretical contribution and for the constructive questions.
>
> ## Q1: How does KLA differ conceptually and practically from SSMs and recurrent sequence models?
>
> Recent work (Liu et al., 2024; Yang et al., 2025) has reinterpreted modern sequence models as solving test-time online regression objectives. We place KLA within this framework in Table 4: LA performs a Hebbian update (1-step GD, no preconditioning); Mamba adds decay/forgetting; Gated DeltaNet adds a scalar-weighted delta rule. All use isotropic Frobenius norms with scalar weights.
>
> KLA solves a precision-weighted least squares objective $\lVert z_t - A_t z_{t-1}^+\rVert^2_{\Lambda_t^-} + \lVert y_t - H_t z_t\rVert^2_{\Sigma_{\text{obs}}^{-1}}$, where both weights are matrices. By Ollivier (2018, 2019), this is algebraically identical to **online natural gradient descent with recursive Fisher preconditioning**.
>
> **In one line:** KLA is to existing SSMs what natural gradient is to vanilla gradient descent.
>
> **Table 4:** Online learning objectives for linear RNN models, following Liu et al. (2024). All prior methods use scalar weights and isotropic Frobenius norms, corresponding to approximate 1-step gradient updates. KLA uses precision-weighted Mahalanobis norms ($\|a\|_M^2 \triangleq a^\top M a$) with a recursively evolving metric — algebraically equivalent to online natural gradient with recursive Fisher preconditioning (Ollivier, 2018, 2019).
>
> | Method | Online Learning Objective | Online Update |
> |---|---|---|
> | Linear Attention | $\underbrace{\lVert S_t - S_{t-1}\rVert_F^2 - 2\langle S_t k_t, v_t \rangle}_{\text{Unregularized least squares}}$ | $\underbrace{S_t = S_{t-1} + v_t k_t^\top}_{\text{Hebbian / 1-step GD}}$ |
> | Mamba | $\underbrace{\lVert S_t - A_t S_{t-1}\rVert_F^2 - 2\langle S_t k_t, v_t \rangle}_{\text{Decay-regularized least squares}}$ | $\underbrace{S_t = \Lambda_t S_{t-1} + v_t k_t^\top}_{\text{Decayed Hebbian / 1-step GD (diagonal forgetting)}}$ |
> | Gated DeltaNet | $\underbrace{\lVert S_t - \alpha_t S_{t-1}\rVert_F^2 - 2\langle S_t k_t, \beta_t(v_t - \alpha_t S_{t-1}k_t)\rangle}_{\text{Gated associative recall}}$ | $\underbrace{S_t = S_{t-1}(\alpha_t(I - \beta_t k_t k_t^\top)) + \beta_t v_t k_t^\top}_{\text{Delta rule / 1-step GD on recall error}}$ |
> | **KLA (ours)** | $\underbrace{\lVert z_t - A_t z_{t-1}^+\rVert^2_{\Lambda_t^-} + \lVert y_t - H_t z_t\rVert^2_{\Sigma_{\text{obs}}^{-1}}}_{\text{Precision-weighted least squares}}$ | $\underbrace{z_t^+ = (\Lambda_t^+)^{-1}(\Lambda_t^- A_t z_{t-1}^+ + H_t^\top \Sigma_{\text{obs}}^{-1} y_t)}_{\text{Online natural gradient with recursive Fisher}}$ |
>
> $\Lambda_t^-$ and $\Lambda_t^+$ are the prior and posterior precisions from the Möbius precision recurrence.
>
> | Symbol | Meaning |
> |---|---|
> | $S_t$, $k_t$, $v_t$ | Memory state, key, value |
> | $\alpha_t$, $\beta_t$ | Scalar gates |
> | $\Lambda_t$ | Diagonal state transition (Mamba) |
> | $z_t$, $z_t^-$, $z_t^+$ | Latent state, prior estimate, posterior estimate (KLA) |
> | $y_t$ | Observation (KLA) |
> | $A_t$, $H_t$ | State transition, observation matrices |
> | $\Lambda_t^-$, $\Lambda_t^+$ | Prior / posterior precision |
> | $\Sigma^{\text{obs}}_t$, $P_t$ | Observation / process noise covariances |
>
>
> ## Q2: Connection and differences with architectures using parallel recurrence?
>
> All parallel-trainable SSMs (Mamba, GLA, Gated DeltaNet) use associative scans over linear recurrences. KLA's scan operates over fractional linear (Möbius) transformations — associative but nonlinear. This nonlinear recurrence is strictly more expressive, as demonstrated in Figure 1 (A₅ permutation composition, NC¹-complete), while remaining parallelizable.
>
> | | Softmax Attention | SSMs / GLA | KLA (Ours) |
> |---|---|---|---|
> | Expressivity | Nonlinear | Linear | Fractional linear (Möbius) |
> | Training / Inference | O(T²) / O(T) | O(T) / O(1) | O(T) / O(1) |
> | Scan structure | N/A | Linear | Möbius |
> | Sequence-level uncertainty / 2nd-order info | ✗ | ✗ | ✓ (recursive $\Sigma_t = \Lambda_t^{-1}$) |
> | Principled noise model | N/A | ✗ | ✓ ($P_t$, $\Sigma^{obs}_t$) |
>
> ## Q3: Runtime benchmarks and GPU memory comparisons?
> We refer to Table 3 under response to Q4 Reviewer kbGX for a detailed comparison.
>
> KLA achieves the **lowest peak memory of all architectures** (4,475 MB; 2.2× reduction vs. GPT). The forward pass (46.8 ms) is on par with GPT and GDN; the backward pass is the current bottleneck, placing throughput (0.42×) in a similar regime to GDN (0.52×). Full block-fusion - as in Mamba and GDN - would further close this gap.
>
> Thanks again for the constructive feedback and we welcome further questions.
>
> **References**
>
> [1] Liu, B. et al. (2024). *arXiv:2407.14207*.
>
> [2] Yang, S. et al. (2025).  *arXiv:2412.06464*.
>
> [3] Ollivier, Y. (2018).  *arXiv:1703.00209*.
>
> [4] Ollivier, Y. (2018).  *arXiv:1901.00696*.

---

> > ### Author Rebuttal · Reviewer_jAQH · 2026-03-31
> >
> > The answers are ok for me. I let my score as it is.

---

> > > ### Author Response · Authors · 2026-04-07
> > >
> > > We thank the reviewer for confirming that our rebuttal addressed the raised concerns. We would like to note that the revised manuscript now includes improved exposition of KLA's conceptual differences from SSMs (Table 4 in the rebuttal, now incorporated), as well as runtime/memory comparisons that were previously missing. We hope these additions also address the presentation and originality concerns flagged in the sub-scores.
> > >
> > > We thank all three reviewers for the constructive engagement during the review process. The manuscript has been revised incorporating pretraining results (45M and 180M scale), hybrid architecture comparisons, and runtime/memory benchmarks with the new CUDA kernel. We are happy that we have been able to address the concerns raised during review, and indeed feel the feedback has helped significantly strengthen the paper.
> > >
> > > Given the revisions and our responses, we would kindly ask each reviewer to consider whether their updated assessment warrants a score update. We are grateful for the feedback that helped improve this work.

---

### Official Review · Reviewer_fJZt · 2026-03-11

**Soundness:** 3
**Presentation:** 3
**Significance:** 3
**Originality:** 3
**Overall Recommendation:** 4
**Confidence:** 4

**Summary:**

The authors propose KLA which reformulates sequence modeling as Bayesian filtering. Each token is treated as a noisy observation of a latent state, and the posterior is maintained as a Gaussian with explicit precision. The key insight is that the Kalman update in information form is a Möbius transformation, which composes associatively and can be computed via parallel prefix scan. This gives nonlinear, history-dependent gating without sacrificing linear-time complexity. KLA is evaluated as a drop-in replacement for SSM/attention layers on synthetic NLP benchmarks, where it matches or outperforms Mamba and GLA, particularly on tasks requiring noise robustness and state tracking.

**Compliance With Llm Reviewing Policy:**

Affirmed.

**Final Justification:**

The paper provides a novel layer, and the rebuttal answered my initial concerns, however, I would like to see the method applied at scale (most easily through hybridization and distillation) I would like to update my score.

**Key Questions For Authors:**

- Is there a reason for why a Gaussian filter is the right inductive bias for discrete token sequences. Since tokens are discrete and highly non-linear.
- How would the proposed layer work with hybrid architectures? GDN based hybrid models are being used effectively in Qwen models, so will using the proposed layer further improve the performance?
- What could be a meaningful way to use the uncertainty parameter for different applications? It appears to be a strength compared to the existing methods, but seems to be unused effectively.
- Do you think the stochasticity of the layers be useful for privacy applications? Can it be used to reduce memorization?

**Limitations:**

Majority are covered in weaknesses but will highlight the key one:
- Missing discussion with relevant works and absence of comparison against attention baselines.
- Authors addressed this but the entire block is not fused in triton.
- The benefit of using this layer compared to GDN/GKA in large models is still unclear.

**Strengths And Weaknesses:**

Strengths:

- The paper is theoretically well grounded with theorems to support the claims.
- The parallelization in the paper is exact which is useful during training, and the non linear gating nicely emerges from bayesian update structure rather than forced by hand.
- The presense of stochasticity in the model could be useful in privacy applications, for instance differential privacy.
- In general the method generalizes over the SSM layers.

Weaknesses:

- Unfortunately, the experiments are only on synthetic tasks, so extrapolating the gains to practical use cases is difficult. It also raises the standard question, when we stack multiple such layers and train with trillions of tokens, will the details of the layer create a significant difference? For instance going from Attention to KLA brings what benefits?
- There is no comparison with attention layer which is the workhouse for all the current generation of models.
- Relevant baseline is not discussed and compared against: https://arxiv.org/pdf/2511.21016
- The posterior uncertainty is not used for any application, like does it help reduce hallucination or generalization?

---

> ### Author Rebuttal · Authors · 2026-03-31
>
> We thank the reviewer for their valuable suggestions and for recognizing the soundness and originality of the framework. We address the concerns below.
>
> ## W1/2: Experiments only synthetic tasks; comparison with attention.. will the details of the layer matter?
>
> We address this on two levels: qualitative expressivity and quantitative pretraining.
>
> **Qualitative**. Figure 1 in the paper evaluates the $A_5$ permutation composition task -  an NC¹-complete problem (Merrill et al., 2024). Mamba and Transformers require depth that grows with sequence length (~4 layers at length 20) to solve this. KLA solves it in constant depth (2 layers) - the only parallel-trainable architecture to do so. This is a qualitative capability separation from **non-linear recurrence**, not an incremental improvement.
>
> **Quantitative**. We have conducted pretraining at 45M and 180M scales comparing Softmax Attention (GPT), Mamba, GDN, standalone KLA, and GPT+KLA hybrid (see Tables in Q1, Reviewer kbGX). Standalone KLA is competitive with all baselines.
>
> ## Q1: Is a Gaussian filter the right inductive bias?
>
> Thank you - this is a genuine doubt. We answer from two perspectives.
>
> **The online learning perspective.** Recent work (see Table under R3-Q1) has reinterpreted modern SSMs/linear attention as solving test-time online regression objectives. LA, Mamba, and GDN all solve online least squares variants with scalar weighting - approximating 1-step gradient updates. KLA solves a **precision-weighted least squares** objective with matrix-valued weights. Under second-moment assumptions alone (Kalman, 1960), this yields the Best Linear Unbiased Estimator (BLUE), without requiring Gaussianity. Under Gaussian assumptions, KLA is additionally the Bayes-optimal estimator.
>
> **Gaussianity is sufficient, not necessary.** Kalman (1960) assumed only: (i) zero-mean noise, (ii) known second moments Q, R, (iii) uncorrelatedness - no distributional form. In KLA, the information-form recurrence $\Lambda_t^+ = \Lambda_t^- + H^\top R^{-1} H$ can be thought of as accumulating Fisher information for natural gradients via amortization and is valid without invoking Bayes' theorem. All existing models operate in continuous states - KLA additionally propagates second-order statistics.
>
> ## Q2: How would KLA work with hybrid architectures?
>
> We replaced the final layer of a 12-layer 180M GPT with an SSM. All three hybrids improve over pure GPT, with GPT+KLA outperforming every baseline on average.
>
> | Model | LAMB↑ | HellaS↑ | PIQA↑ | Arc-E↑ | Arc-C↑ | WinoGr↑ | OBQA↑ | BoolQ↑ | Avg↑ |
> |---|---|---|---|---|---|---|---|---|---|
> | GPT+Mamba | 26.60 | 31.60 | 59.80 | 52.60† | 24.80† | **54.40** | 31.20† | 57.00† | 42.25 |
> | GPT+GDN | 29.80† | **35.00** | **64.20** | 54.00 | 25.20 | 49.40 | **32.20** | 56.40 | 43.28† |
> | GPT+KLA ⭐ | **30.20** | 34.40† | **64.20** | **55.80** | **29.00** | 51.80† | 31.00 | **63.20** | **44.95** |
> > **Bold** = best. † = second best.
>
> ## W4/Q3: Meaningful use of uncertainty?
>
> KLA's gating is driven entirely by uncertainties(Eq. 14-15, Figure 2) - unlike Mamba where gating emerges from a token-dependent discretization parameter $\Delta_t$. We deliberately kept KLA dynamics time-invariant (even though its more expressive) to isolate this. We further show the Monte Carlo variant (KLA+, Table 2 main) when explicitly passes gradients through the precisions, further improves performance. Extending this to sampling / variational inference based at scale (large vocabularies, principled precision propagation across blocks) is non-trivial and a key direction for future work.
>
> ## Q4: Can stochasticity help with privacy?
>
> KLA's gate structure (Eq. 14–15) with noise parameters provides a natural interface for privacy and introducing DP-noise. Observation precision $\Lambda_t^{\text{v}}$ controls the input gate, bounding per-token sensitivity. The forget gate is controlled by process noise $p_t^-$. The ratio between these directly governs a privacy-utility frontier. A formal DP-KLA analysis is compelling future work.
>
> ## W3: Missing baseline
>
> Thank you for pointing out this insightful concurrent work (arXiv:2511.21016). We have cited this in the revised manuscript.
>
> GKA's starting premise, stated explicitly in the paper, is that a general KF's "sequential dependency prevents the parallelization necessary for ... modern hardware." GKA hence adopt a static deterministic dynamics assumption (**A_t = I, no process noise**) for simplification, which is in fact a linear recurrence.
>
> Our fundamental contribution is showing that **this sequential dependency can be parallelized exactly** via information-form reparameterization - no steady-state assumption, no iterative approximation.
>
> To quantify the impact of these assumptions, we ablate process noise in KLA (Figure 9). Removing it causes the Möbius structure to become a linear recurrence, and performance degrades.
>
> We hope these results address the reviewer's concerns and welcome further questions.

---

> > ### Author Rebuttal · Reviewer_fJZt · 2026-04-03
> >
> > I thank the authors for the extra experiments, and would like to update my score.

---

> > > ### Author Response · Authors · 2026-04-07
> > >
> > > We sincerely thank the reviewer for the thoughtful re-evaluation and for updating their score. Your feedback on pretraining and hybrid evaluations has meaningfully improved the paper.

---

### Official Review · Reviewer_kbGX · 2026-03-16

**Soundness:** 3
**Presentation:** 4
**Significance:** 3
**Originality:** 3
**Overall Recommendation:** 4
**Confidence:** 3

**Summary:**

This paper proposes Kalman Linear Attention layer, utilizing history dependent non-linear gating mechanism based on precision. The authors reparameterize the Kalman filter in information form, which enables efficient parallel training like Mamba and GLA. Their implementations for Triton kernels shows faster training-time runtime scaling across various sequence lengths. KLA shows good scaling effects and performance on various domains.

**Compliance With Llm Reviewing Policy:**

Affirmed.

**Final Justification:**

Due to unavoidable circumstances as I stated in the official comment, I was unable to participate in the rebuttal discussion, and I sincerely apologize to the authors for this. Nevertheless, the concerns I raised reflect what I considered the most critical issues of this submission. I kindly ask the AC to carefully check whether these concerns have been adequately addressed in the authors' rebuttal. If not, I would maintain my current rating.

**Key Questions For Authors:**

See above the weaknesses sections.

**Limitations:**

See above the weaknesses sections.

**Strengths And Weaknesses:**

Strengths:

- The paper is well-written and organized. It was easy to follow the paper.
- The motivation and the connections to the proposed method are great, updating precision for value and history through Bayesian filtering.
- KLA uses non-linear precision-ratio gates, while enabling parallel training through associative scan.
- The overall experimental results and analyses support the superior or comparable performance compared to the baselines.


Weaknesses:

- Although MAD-lab showed language modeling on their synthetic benchmark can be generalized well to real-world benchmarks, I believe this new architecture should be also evaluated on real pretraining corpus. Are there any results on pretraining? I feel like this task is most important to validate the proposed architecture.
- Could you also describe the FLOPs and parameter sizes of the models including KLA and baselines?
- When I look at hyperparameters in Appendix, the model is using only a few blocks like 1 or 2. Is KLA unstable on larger-scale settings, which prevents the authors from experimenting real pretraining?
- Can you also compare the training-time runtime (in Figure 5) with Mamba or GDN? Triton scan looks good for KLA, but I'm wondering how fast it is compared to other models.

---

> ### Author Rebuttal · Authors · 2026-03-31
>
> We thank the reviewer for the positive reviews and suggestions for improvement. We aim adrress your questions below.
>
> ## Q1: Are there any results on pretraining?
>
> We have conducted pretraining at two scales: **45M** (12 blocks, 1.8B tokens, 2× Chinchilla) and **180M** (12 blocks, 10.9B tokens, 3× Chinchilla) on OpenFineWebEdu with AdamW (weight decay 0.1). We compare against GPT, Mamba, and GDN.
>
> **Table 1: 45M Scale (12 Blocks)**
>
> | Model | LAMB↑ | HellaS↑ | PIQA↑ | Arc-E↑ | Arc-C↑ | WinoGr↑ | OBQA↑ | BoolQ↑ | Avg↑ |
> |---|---|---|---|---|---|---|---|---|---|
> | GPT | **16.00** | 28.60 | 58.00 | 40.20† | 20.60 | 50.00 | **27.40** | 37.40 | 34.78 |
> | Mamba | 9.20† | **29.20** | **59.80** | 37.20 | **25.40** | 48.80 | 25.40 | 61.60† | 37.08 |
> | GDN | **16.00** | 28.20 | 58.40 | 39.00 | 23.40† | **52.60** | 27.20 | 45.20 | 36.25 |
> | **KLA (Ours)** | 7.60 | 28.80† | 58.60† | **40.60** | 23.20 | 52.20† | **27.40** | **63.00** | 37.68† |
> | GPT+KLA ⭐ | **16.00** | 28.20 | 58.60† | **42.80** | 23.00 | 50.80 | 24.40 | 62.80† | **38.33** |
>
> > **Bold** = best. † = second best.
>
> ---
>
> **Table 2: 180M Scale (12 Blocks)**
>
> | Model | LAMB↑ | HellaS↑ | PIQA↑ | Arc-E↑ | Arc-C↑ | WinoGr↑ | OBQA↑ | BoolQ↑ | Avg↑ |
> |---|---|---|---|---|---|---|---|---|---|
> | GPT | **28.80** | 32.20† | 63.40† | **51.00** | 27.00† | 49.20 | 31.20 | 54.80 | 42.20 |
> | Mamba | 24.00 | 30.60 | 63.00 | **51.00** | **27.20** | **53.80** | **33.00** | 55.80 | 42.30† |
> | GDN | 27.80† | 31.60 | 60.40 | 45.60 | 25.80 | 53.40 | 30.00 | **60.80** | 41.93 |
> | **KLA (Ours)** | 22.40 | **34.00** | 63.00 | 50.00† | 26.00 | 53.60† | 32.40† | **60.80** | **42.78** |
> | GPT+KLA ⭐ | **30.20** | **34.40** | **64.20** | **55.80** | **29.00** | 51.80 | 31.00 | 63.20† | **44.95** |
>
> > **Bold** = best. † = second best.
>
> At both scales, on commonsense reasoning benchmarks KLA matches or outperforms baselines. Interestingly, GPT+KLA hybrid (with a KLA block as the final layer) improves upon GPT getting the best of both worlds. Further hybrid results are under Q2, R-fJZt.
>
> At 24 layers, other baselines narrow the gap, as increased depth compensates for their architectural limitations at shallower configurations - consistent with Figure 1 in the paper. Full 24L tables are in the updated manuscript along with all hyperparameters.
>
>
> ## Q2/4: Could you describe the FLOPs and parameter sizes, compare training-time runtime with Mamba or GDN?
>
> We now provide a direct comparison at 45M scale (BS=4, seq_len=1024, A100 80GB):
>
> **Table 3: Training-time runtime and memory at 45M scale**
>
> | Config | Params | Fwd (ms) | Bwd (ms) | Total (ms) | Peak (MB) | Speedup |
> |---|---|---|---|---|---|---|
> | GPT-45M | 45.0M | 42.2 | 56.7 | 98.9 | 9,965 | 1.00× |
> | Mamba-45M | 45.5M | 21.7 | 44.9 | 66.6 | 5,762 | 1.49× |
> | GDN-45M | 43.9M | 44.1 | 147.0 | 191.2 | 6,690 | 0.52× |
> | KLA (Torch scan) | 45.3M | _OOM_ | _OOM_ | _OOM_ | _OOM_ | _N/A_ |
> | KLA (Triton scan) | 45.3M | 155.0 | 381.8 | 536.6 | 11,080 | 0.18× |
> | **KLA (CUDA kernel)** | 45.3M | 46.8 | 186.2 | 233.1 | **4,475** | 0.42× |
>
> At submission time, Figure 5 compared only KLA's own implementations. The Torch-based scan (`torch._higher_order_ops.associative_scan`) ran out of memory. The Triton kernel remained in a memory-bound regime (11,080 MB, 0.18× throughput).
>
> Post-submission, we developed a fused CUDA kernel that performs the Möbius associative scan in shared memory, avoiding materialization of intermediate tensors in HBM. This achieves **peak memory of 4,475 MB — the lowest of all architectures** (2.2× reduction vs GPT, 1.3× vs Mamba), with a forward pass (46.8 ms) comparable to GPT / GDN. The backward pass (186.2 ms) remains the bottleneck, yielding 0.42× total throughput - within the same regime as GDN (0.52×). Fusing the full block into a single kernel is a natural next step that would further close this gap.
>
> ## Q3: Is KLA unstable on larger-scale settings (only 1-2 blocks in Appendix)?
>
> No, KLA scales stably to deep architectures. Our 45M model uses **12 stacked KLA blocks** and our 180M model uses **12 and 24 stacked KLA blocks**, with no gradient explosions or instabilities (gradient clipping factor 3.0 for all models).
>
> The key to stability is the mean-reverting Ornstein–Uhlenbeck (OU) process prior (Figure 2), which we consider a major contribution. Early in development, naively stacking probabilistic modules did produce instabilities. Standard engineering fixes (gradient clipping, lower learning rates) didn't resolve the root cause. The OU discretization addresses it principally by ensuring latent dynamics are bounded - enabling, to our knowledge for the first time, stable backpropagation through architectures that propagate distributions between blocks and through time.
>
> We currently propagate only posterior means across blocks, which proved sufficient for competitive performance. Propagating full posteriors is a natural extension, a design choice for richer expressivity, not a stability requirement.

---

### Decision · Program_Chairs · 2026-04-30

**Decision:**

Accept (regular)

**Comment:**

This paper introduces the Kalman Linear Attention (KLA) layer which offers an efficient alternative to standard attention layers by reformulating sequence modeling as Bayesian filtering. The reviewers agreed that the approach is well-motivated and offers a novel theoretical contribution grounded in a principled probabilistic framework. Experimental results support the efficiency and performance claims. Moreover, the paper is well-written and clear. Reviewers initially raised concerns about whether the method would work at scale on a large pre-training corpus. The authors addressed these concerns during the rebuttal phase through new pre-training results showing comparable or better performance relative to GPT, Mamba, and GDN baselines. The proposed KLA layer is highly relevant and claims are well-supported by theory and experiments.